# Atmospheric deposition and river runoff stimulate the utilization of dissolved organic phosphorus in coastal seas

Haoyu Jin[1,2,3], Chao Zhang [1,2] ✉, Siyu Meng[1,2,3], Qin Wang[1,2], Xiaokun Ding[4], Ling Meng[5,6], Yunyun Zhuang[1,2], Xiaohong Yao [1,2], Yang Gao [1,2], Feng Shi[1,2], Thomas Mock [3] ✉ & Huiwang Gao [1,2] ✉

In coastal seas, the role of atmospheric deposition and river runoff in dissolved organic phosphorus (DOP) utilization is not well understood. Here, we address this knowledge gap by combining microcosm experiments with a global approach considering the relationship between the activity of alkaline phosphatases and changes in phytoplankton biomass in relation to the concentration of dissolved inorganic phosphorus (DIP). Our results suggest that the addition of aerosols and riverine water stimulate the biological utilization of DOP in coastal seas primarily by depleting DIP due to increasing nitrogen concentrations, which enhances phytoplankton growth. This "Anthropogenic Nitrogen Pump" was therefore identified to make DOP an important source of phosphorus for phytoplankton in coastal seas but only when the ratio of chlorophyll $a$ to DIP [$\log_{10}$ (Chl $a$ / DIP)] is larger than 1.20. Our study therefore suggests that anthropogenic nitrogen input might contribute to the phosphorus cycle in coastal seas.

Phosphorus (P) is an essential element for microorganisms such as phytoplankton due to its relevance for the synthesis of nucleic acids, phospholipids, and the main energy currency in cells, adenosine 5′-triphosphate[1,2]. Dissolved inorganic phosphorus (DIP) is regarded as the main bioavailable phosphorus and its concentration is often low in the surface open oceans with the exception of coastal seas[3,4]. Low DIP concentrations have been widely observed in open oceans such as the Atlantic and the Pacific Oceans, and sometimes also in coastal seas[4–6]. In contrast, dissolved organic phosphorus (DOP) in the oceans is more abundant relative to DIP[7,8]. Thus, microbes including phytoplankton have evolved to efficiently utilize DOP through extracellular hydrolysis, which converts DOP into DIP for cellular uptake[5,9] and therefore mitigates phosphorus limitation under conditions of DIP scarcity.

The most important DOP hydrolase is the alkaline phosphatase (AP), which can be active in different subcellular compartments such as the extracellular space (e.g., phycosphere), the cell wall, the plasma membrane, or as part of the cytoplasm of diverse microbes including phytoplankton[1,10,11]. The activity of AP (APA) is mainly composed of inducible APA and constitutive APA. Although there is evidence of background APA at relatively low levels (constitutive APA) independent of DIP concentrations[12–14], AP usually is induced by low concentrations of DIP. Thus, the APA usually is inversely correlated with DIP concentrations, and inducible AP therefore becomes activated for the utilization of DOP when DIP concentrations fall below a certain threshold[15–17]. In addition to the DIP concentration, other factors such as the abundance and diversity of microbiomes, N:P ratios, seawater temperature, and the availability of essential enzyme co-factors such

[1]Frontiers Science Center for Deep Ocean Multispheres and Earth System, and Key Laboratory of Marine Environment and Ecology, Ministry of Education of China, Ocean University of China, Qingdao 266100, China. [2]Marine Ecology and Environmental Science Laboratory, Laoshan Laboratory, Qingdao 266071, China. [3]School of Environmental Sciences, University of East Anglia, Norwich Research Park, Norwich NR4 7TJ, UK. [4]School of Ocean, Yantai University, Yantai 264005, China. [5]CAS Key Laboratory of Coastal Environmental Processes and Ecological Remediation, Yantai Institute of Coastal Zone Research, Chinese Academy of Sciences, Yantai 264003, China. [6]University of Chinese Academy of Sciences, Beijing 100049, China. ✉e-mail: zhangchao@ouc.edu.cn; T.Mock@uea.ac.uk; hwgao@ouc.edu.cn

as Fe and Zn also affect APA[11,18–20]. Hence, these variables might therefore have an impact on the DIP threshold for the activation of AP as previously shown[11,13,21]. This complex system of variables impacting the activation of AP likely is the reason why there is no consensus yet in terms of environmental conditions that induce DOP utilization in the ocean and especially in coastal seas.

Atmospheric deposition and river runoff are considered important sources of nutrients[22–25] for the surface ocean as they usually contain significant amounts of nitrogen (N) and trace metals (e.g., Fe) but relatively low concentrations of P[26–30]. Studies have shown that atmospheric deposition can contribute between 40% and 70% of the total terrigenous N input in the North Pacific, North Atlantic, and Mediterranean[31–33]. Similar, riverine input can contribute 19–37 Tg N year$^{-1}$ to the global coastal seas and is considered an important reason leading to coastal eutrophication[24,34]. Furthermore, atmospheric and riverine inputs are known to be indispensable sources of Fe in the global ocean[26,30]. Hence, atmospheric deposition has been estimated to stimulate phytoplankton growth and therefore to support between 3% and 5% of global marine net primary productivity[35,36], while river runoff, of which the impact is mainly concentrated in the coastal waters, has been estimated to enhance the net primary productivity by ~14% in the global coastal seas over the 1905–2010 period[34]. Atmospheric deposition can also supply nutrients to the deeper ocean where it supports phytoplankton growth in the deep chlorophyll *a* maximum layer (DCM)[37,38]. In the recent past, the N input through atmospheric deposition and river runoff has increased due to increasing economic activities[27,34]. This might lead to an excess of N especially in seawater closest to the source of anthropogenic pollution[39–41]. Hence, coastal seas are likely more affected than the open ocean in terms of an increased input of N, which can lead to elevated N:P ratios[24,42] and therefore potentially a gradual shift from N deficiency to P deficiency[43]. Under such biogeochemical conditions, DOP might become an important source of P for phytoplankton to alleviate limitation by DIP in excess of N[1,5,44]. However, despite decades of research on the impacts of atmospheric deposition and riverine input of essential nutrients on marine biogeochemistry[28,34,36,45], their role in DOP utilization in coastal seas remains elusive.

Thus, only a few studies so far have focused on meso-/eutrophic coastal waters when it comes to the role of atmospheric deposition and riverine input in regulating DOP utilization[18,46,47]. Most studies have focused on oligotrophic open oceans where only atmospheric deposition can be considered because the geographical reach of rivers is more limited due to significant mixing and advection in coastal seas. Atmospheric deposition in oligotrophic open oceans can enhance DOP utilization and therefore alleviate P stress, but primarily by supplying essential enzyme co-factors of AP such as Fe and Zn[11,17]. In contrast, our recent work has provided preliminary evidence that atmospheric deposition can stimulate phytoplankton growth in the China Coastal Seas even though the concentrations of DIP in this geographical area were unable to meet the cellular requirements of phytoplankton for growth[46,47].

Westerly winds in East Asia can carry aerosols into the China Coastal Seas, including the Bohai Sea, the Yellow Sea, and the East China Sea[48–50] and therefore significantly contribute to atmospheric deposition in these coastal seas. Furthermore, multiple rivers such as the Yellow River, the Xiaoqing River and the Yangtze River significantly contribute to riverine input of terrestrial nutrients in these coastal seas. In the relatively enclosed Bohai Sea, both atmospheric deposition and river runoff cause excess macro-nutrient concentrations (e.g., dissolved inorganic nitrogen, DIN) and elevated concentrations of trace metals[51,52]. Consequently, P has now become the main limiting nutrient[53–55]. The Yellow Sea and the East China Sea are marginal seas connected to open waters of the Northwest Pacific Ocean, with typical biogeochemistry spanning from eutrophy to oligotrophy with increasing distance from the East Asian continent[56]. In contrast to the

increased anthropogenic input of N, DIP concentrations have been continuously decreasing since the mid-1990s, hence causing P deficiency[43,57]. Thus, these coastal seas provide an ideal test bed for studying the effect of atmospheric deposition and river runoff on the utilization of DOP in relatively nutrient-rich coastal waters, which contribute disproportionally to global biogeochemical cycles[58,59].

To address this knowledge gap, we carried out a series of on-board microcosm experiments in the China Coastal Seas covering a gradient from eutrophic to oligotrophic conditions with the aim to reveal the impact of atmospheric deposition and riverine input on DOP utilization and therefore the role of N-rich aerosols and riverine waters in biogeochemical cycles of coastal seas (Fig. 1).

## Results

### Characteristics of atmospheric aerosols, riverine water, and seawater of the China Coastal Seas

The concentration of DIN (0.99–2.14 μmol·m$^{-3}$) in aerosols was much higher than that of DIP (0.66 × 10$^{-3}$–8.36 × 10$^{-3}$ μmol·m$^{-3}$), resulting in high DIN:DIP ratios (>100), which significantly exceeded the Redfield ratio (N:P = 16:1) (Table S1). Similarly, the concentration of DIN in the riverine water was high (330.54 μM) and that of DIP was low (0.65 μM), resulting in a high N:P ratio of 509 (Table S1). In addition, the salinity of the 20 L incubation system decreased about 0.05 PSU for the River-low group and 0.2 PSU for the River-high group, which had negligible impacts on phytoplankton growth[60]. The DIN:DIP ratios were also high (>100) in the seawater at B1$_{Spr}$ (127) and B1$_{Sum}$ (107) in the Bohai Sea, E1 (151) in the East China Sea, and Y1 (200) in the Yellow Sea, while closer to the Redfield ratio at Y3$_{Spr}$ (12), Y1-DCM (18), and Y2 (15) in the Yellow Sea.

Generally, seawater in this study was characterized by low DIP concentrations (0.01–0.11 μM) with the highest value measured at the DCM layer. In contrast, the DIN (0.23–9.83 μM), Si(OH)$_4$ (0.20–8.47 μM), and Chl *a* (0.14–4.47 μg·L$^{-1}$) concentrations varied widely (Table S2). However, we were able to divide the nutrient status of the seawater used in this study for the microcosm experiments into three categories: I. Eutrophic-like: At B1$_{Sum}$ and DCM layer of Y1, Chl *a* (1.21–2.94 μg·L$^{-1}$), DIN (1.99–3.60 μM), and Si(OH)$_4$ (3.10–8.47 μM) concentrations were relatively high. At Y3$_{Spr}$ and B1$_{Spr}$, the Chl *a* concentrations were also high (3.07–4.47 μg·L$^{-1}$), even though DIN (<1 μM) and Si(OH)$_4$ (0.20–0.79 μM) concentrations were relatively low. This is likely because the seawater was collected during a spring bloom. In contrast, DIN (9.83 μM) and Si(OH)$_4$ (3.18 μM) concentrations were high, while the Chl *a* concentration was relatively low at E1(0.43 μg·L$^{-1}$), which corresponded to the end of the spring bloom. II. Mesotrophic-like: The seawater of Y1 and Y3$_{Sum}$ had a medium level of Chl *a* (0.47–1.16 μg·L$^{-1}$), DIN (0.23–1.53 μM), and Si(OH)$_4$ (1.04–1.69 μM) concentrations. III. Oligotrophic-like: The seawater of Y2 had low nutrient concentrations (DIN, 0.27 μM; DIP, 0.02 μM; Si(OH)$_4$, 1.02 μM), and lowest Chl *a* concentration (0.14 μg·L$^{-1}$) (Table S2).

### The response of phytoplankton to additions of aerosols and riverine water

Even though the in-situ Chl *a* and nutrient concentrations in the sampled seawater varied over an order of magnitude at different stations (Table S2), a significant response of Chl *a* concentrations to the addition of nutrients, aerosols, and riverine waters was found at most stations (Figs. S1 and S2). However, considering that seawater samples were collected from different nutrient regimes (eutrophic to oligotrophic) and phytoplankton growth stages (Figs. S1, S2, and Table S2), it was challenging to delineate the response of Chl *a* under different experimental treatments conducted with the microcosms. ΔR, which has been used to quantify the relative change of Chl *a* after diverse treatments in previous studies[61], was therefore introduced to determine the significance of the response of Chl *a* to additions of nutrients,

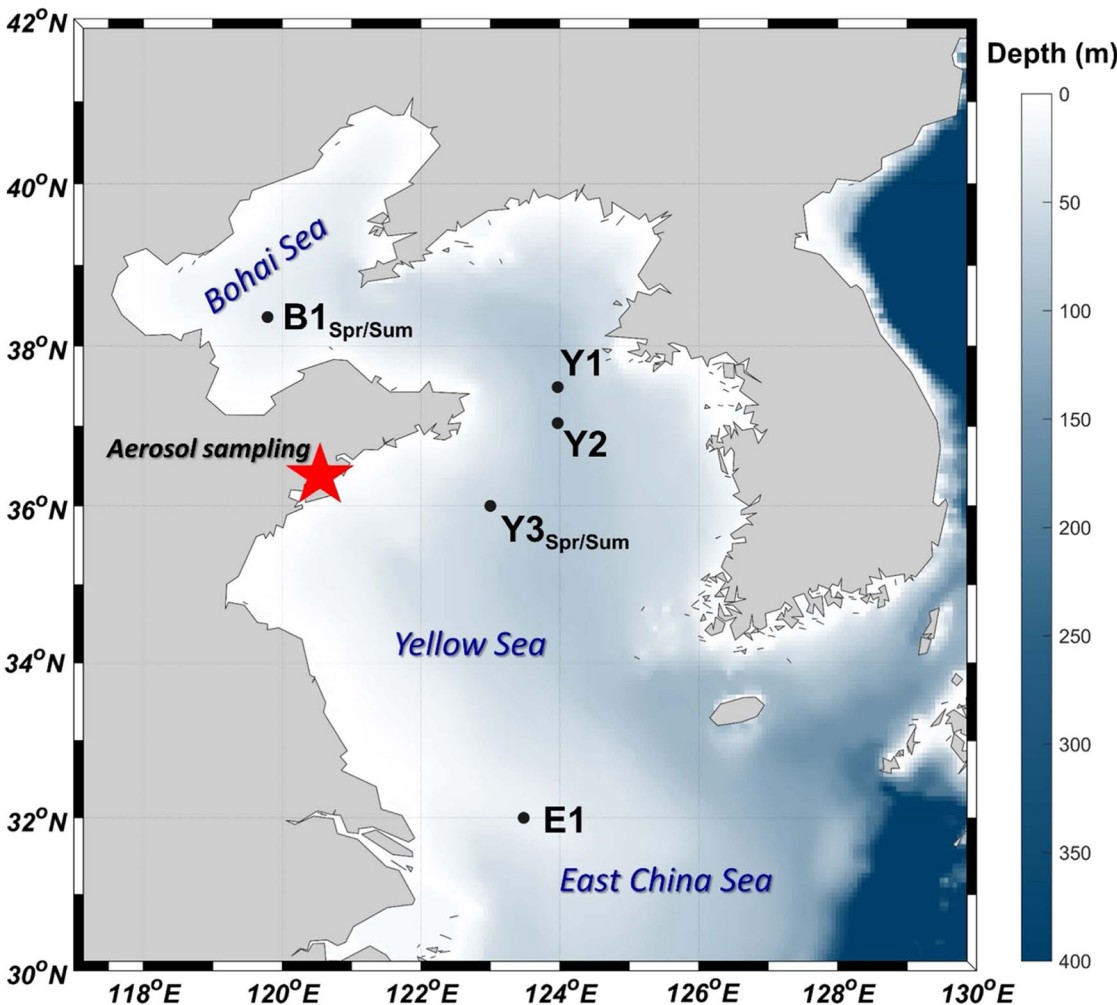

**Fig. 1 | Aerosol and seawater sampling stations for the microcosm experiments.** The experiments in spring and summer at Y3 (B1) are labeled as $Y3_{Spr}$ ($B1_{Spr}$) and $Y3_{Sum}$ ($B1_{Sum}$), respectively. The red star indicates the location of aerosol sampling. Source data are provided as a Source Data file.

aerosols, and riverine waters. According to our $\Delta R$ calculations for nutrient treatments, phytoplankton was generally limited by N in spring, and by either P or N + P in summer (Fig. 2a). Note that phytoplankton at E1 and Y1-DCM were not growth limited by either N or P (Fig. 2a). However, additions of aerosols or riverine waters appear to have increased Chl $a$ and $\Delta R$ in almost all experiments ($p < 0.05$) irrespective of the limiting nutrient, and the addition of aerosols and riverine waters with same DIN content shows a similar increase of Chl $a$ (Figs. 2b and S2). Even under the limitations of P and N + P, the additions of aerosols or riverine waters with minimal DIP supply relative to the concentrations in natural seawater, increased Chl $a$ concentrations over time (Figs. 2, S2, and S3).

## The activity of alkaline phosphatases and DOP utilization in microcosm experiments

After the addition of aerosols or riverine waters, the DIN and DIP concentrations decreased in most incubations (Fig. S3). The APA in the surface seawater varied greatly during spring- and summertime. Low APAs were observed at E1 (0.4 nM P·h$^{-1}$) and $B1_{Spr}$ (0.9 nM P·h$^{-1}$) in spring with lower seawater temperature (10–17 °C), whereas much higher (9.0–179.1 nM P·h$^{-1}$) APA was measured in the summer with higher seawater temperature (28–32 °C), due to the increased severity of P deficiency except for Y1-DCM (2.1 nM P·h$^{-1}$) that had relatively high DIP concentrations (Table S2). This seasonal difference was most pronounced at B1 in the Bohai Sea, of which the APA in summer was

more than two orders of magnitude higher than in spring (Table S2). The APA generally showed a positive response to the addition of DIN (Figs. S4a, c and S5a-b), aerosols (Fig. 3) and riverine waters (Fig. S6a, b) whereas it decreased after the addition of P or N + P (Fig. S5). A significant increase of Chl $a$ normalized APA (APA / Chl $a$) was also found at $B1_{Spr}$, $B1_{Sum}$, and $Y3_{Sum}$ (Fig. S7).

Compared with concentrations in surface seawater, The concentration of total dissolved phosphorus (TDP) changed by -42.5%–5.9% during the incubations (Fig. S8), indicative of substantial concentrations of P in seawater. The net utilization of DIP and DOP by phytoplankton were estimated through the change of DIP ($\Delta$DIP) and DOP ($\Delta$DOP), respectively, during the incubations. There were three patterns observed with respect to the utilization of P:

I. DIP was the main source of phosphorus. In experiments using eutrophic seawater with relatively high DIP concentrations (0.07–0.11 μM) including E1 and Y1-DCM, DIP was the main bioavailable P nutrient, and 0.04–0.05 μM DIP was consumed by phytoplankton in the control groups (Fig. 4a and S9a, f). The DIP utilization changed by -5.5%–91.0% after aerosol addition, which was positively correlated with the added amount at Y1-DCM (Fig. S9f).

II. DIP was the main source of phosphorus in the control and DOP was the main source of phosphorus in the aerosol treatments. DIP was also the main P nutrient in the control at $Y3_{Spr}$ and Y2

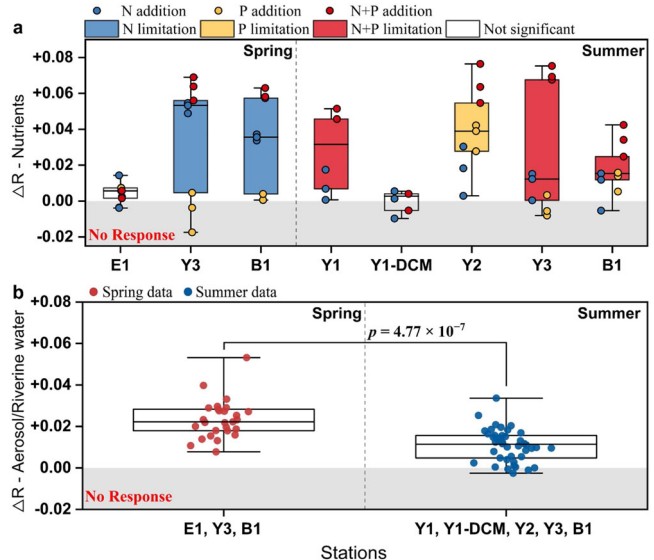

**Fig. 2 | Response of Chlorophyll *a* (Chl *a*) concentration to nutrient, aerosol and riverine water additions expressed by ΔR.** ΔR ($\log_{10}$(Chl $a_{\text{t-avg}}$/Chl $a_{\text{c-avg}}$)/$t$) is the $\text{Log}_{10}$ ratio of the average Chl *a* concentration during the incubations in treatment relative to control groups. **a** ΔR in nutrient addition experiments. The colored dots indicate ΔR in different treatment groups ($n$ = 3 samples). The color in boxes indicates the significant response of Chl *a* to treatments (limiting nutrients) calculated using one-way ANOVA and Turkey HSD post hoc test (blue = N limitation, yellow = P limitation, red = N + P colimitation, and white = not significant) ($p < 0.05$). The lines across the boxes represent $25^{\text{th}}$ (bottom), $50^{\text{th}}$ (middle), and $75^{\text{th}}$ (top) percentiles. The whisker caps represent minimum (bottom) and maximum (top) values. **b** ΔR in aerosol and riverine water addition experiments. The boxes with blue and red dots indicate the ΔR in spring and summer, respectively. The lines across the boxes represent $25^{\text{th}}$ (bottom), $50^{\text{th}}$ (middle), and $75^{\text{th}}$ (top) percentiles. The whisker caps represent minimum (bottom) and maximum (top) values. Significant differences ($p < 0.05$) between ΔR in spring and summer are indicated with a connecting line and $p$ value (unpaired t-test, two-sided). $n$, number of samples. Source data are provided as a Source Data file.

(Fig. S9b, g). Except for Y1-DCM which had higher DIP, DOP utilization was induced after the aerosol addition, leading to small uptake ($\leq 0.02\,\mu M$) of DOP during the incubations in the aerosol treatments at E1, Y3$_{\text{Spr}}$, and Y2. Aerosol addition contributed to DOP becoming the main P nutrient for phytoplankton growth, which accounted for 51.9%–59.7% of P uptake in the high-content aerosol treatment at Y3$_{\text{Spr}}$ and in the low- and medium-content aerosol treatments at Y2 (Fig. 4b and S9a, b, g).

III. DOP was the main source of phosphorus. At B1$_{\text{Spr}}$, B1$_{\text{Sum}}$, Y1, and Y3$_{\text{Sum}}$, where DIP was highly deficient (0.01–0.03 μM), DOP contributed 82.4%–92.5% of P and became the primary P nutrient for phytoplankton growth in the control groups (Fig. 4c). However, although DOP was still the main P nutrient in the treatment groups, no significant enhancement of ΔDOP was found after aerosol additions (Fig. S9c, d, e, h).

Note that ΔDOP as net DOP utilization did not show a significant response either to aerosol or riverine water additions in most experiments (Fig. 4, S6c-d, and S9), which was likely ascribed to the high turnover rate of easily degradable DOP in coastal seas[62,63]. We therefore defined ΔDOP* (ΔDIN / 16−ΔDIP, Eq. 3) on the basis of DIN (ΔDIN) and DIP (ΔDIP) utilization (Eq. 2), and the Redfield ratio for N:P requirement (16) of phytoplankton. This enabled us to estimate the potential maximum uptake of DOP (versus net uptake given as ΔDOP[11,44], which was positively correlated with ΔDOP and APA (Fig. 5a). At E1 and Y1-DCM, where DIP was the main P nutrient, ΔDOP* was low (-0.06–0.02 μM) (Fig. 4a). At Y3$_{\text{Spr}}$ and B1$_{\text{Spr}}$ with ΔDOP

between -0.02 and 0.02 μM, positive ΔDOP* could be found. Moreover, at B1$_{\text{Sum}}$ and Y3$_{\text{Sum}}$ with high APA (15.5–513.7 nM P·h⁻¹) and ΔDOP between 0.04–0.06 μM, phytoplankton had a larger potential to take up DOP (ΔDOP*, up to 0.29 μM), indicating that the total uptake of DOP during the incubations may be up to 37 times higher than the calculated net DOP uptake (ΔDOP, up to 0.06 μM) (Fig. 4, S6, and S9). In addition, ΔDOP* significantly responded to aerosol and riverine water additions at Y3$_{\text{Spr}}$, B1$_{\text{Spr}}$, Y2, Y3$_{\text{Sum}}$, and B1$_{\text{Sum}}$ where DOP was the main P nutrient or might have become the main P nutrient after aerosol additions, of which the response was positively correlated with the amount added (Fig. 4b-c, S6, and S9). ΔDOP* at B1$_{\text{Sum}}$ in the Bohai Sea was even higher than the DOP concentration at the end of the incubations, which is in accordance with an increased turnover of DOP in the treatments with aerosol and riverine water (Figs. 4, S6, and S9). As an exception, although APA was high at Y1 (26.2–121.0 nM P·h⁻¹), ΔDOP* did not significantly respond to aerosol additions. This is possibly due to the contribution of bacteria to DOP hydrolysis, as the Chl *a* normalized APA (APA / Chl *a*) at Y1 did not significantly respond to aerosol addition, and was much higher than that of other stations with significant DOP utilization (B1$_{\text{Spr}}$, Y3$_{\text{Sum}}$, B1$_{\text{Sum}}$) (Figs. S7 and S9).

It should be noted that although phytoplankton can alleviate P limitation by increasing the N:P uptake ratio under P-limited conditions[64], the higher ΔDOP* relative to ΔDIP was also identified using three (ΔDOP*48 = ΔDIN / 48−ΔDIP, generally indicative of obvious P deficiency) and even if five (ΔDOP*80 = ΔDIN / 80−ΔDIP) times the Redfield ratio (Fig. S10). Thus, our results do not appear to be significantly impacted by differences in the Redfield ratio, which therefore corroborates the robustness of the parameter ΔDOP*.

## Discussion

The APA and therefore the utilization of DOP usually is controlled by various factors such as the DIP concentration, seawater temperature, phytoplankton biomass, and the biodiversity and overall activity of the microbial communities. Furthermore, trace metals used as essential enzyme co-factors (e.g., Fe, Zn) for the activity of AP[1,11,20,44,65] play a key role. However, the DIP concentration is regarded as the most important factor controlling the DOP utilization. DIP is known to be inversely correlated with the APA[1,17]. Previous studies[5,13,16,17,21] have shown that APA increases when the DIP concentration is below a certain threshold, known as the DIP threshold for the APA (Table S3). This DIP threshold has been confirmed by our study through applying a segmented regression approach (Fig. S11). For instance, at stations in the Bohai Sea, the DIP threshold was identified to be at 0.02 μM DIP in the summer (B1$_{\text{Sum}}$) although no threshold was identified in spring (B1$_{\text{Spr}}$) at the same station (Fig. S11b). In the East China Sea and the Yellow Sea, the DIP threshold was 0.02 μM but only when the only oligotrophic station (Y2) was excluded (Fig. S11a). The APA and DIP concentration at Y2 characterized as oligotrophic-like were also close to those measured in oligotrophic open oceans[17,21,66], which corroborates the validity of our data. However, even though the importance of DIP has been well studied, the DIP thresholds generally vary in different oceanic regions (Table S3).

In addition, we found that Chl *a* concentrations were positively correlated with APA, ΔDOP and ΔDOP* (Fig. 5a). Chl *a* normalized APA (APA / Chl *a*) was also significantly increased after aerosol and riverine water additions in the Bohai Sea (B1$_{\text{Spr}}$, B1$_{\text{Sum}}$) and Yellow Sea (Y3$_{\text{Sum}}$) (Fig. S7), which suggests that phytoplankton biomass plays a key role in the DOP utilization in coastal seas[65]. The $\text{Log}_{10}$ ratio of Chl *a* / DIP [$\text{Log}_{10}$ (Chl *a* / DIP)] was therefore introduced as an indicator for the likelihood of elevated P requirement by phytoplankton. A positive correlation was found between $\text{Log}_{10}$ (Chl *a* / DIP) and the relative change of APA after the additions of aerosol and riverine water, indicating DOP utilization as a consequence of aerosol deposition and river runoff in coastal seas (Fig. 6a). In addition, the relative change of APA was not significantly correlated with N* (DIN−16 × DIP) in terms of

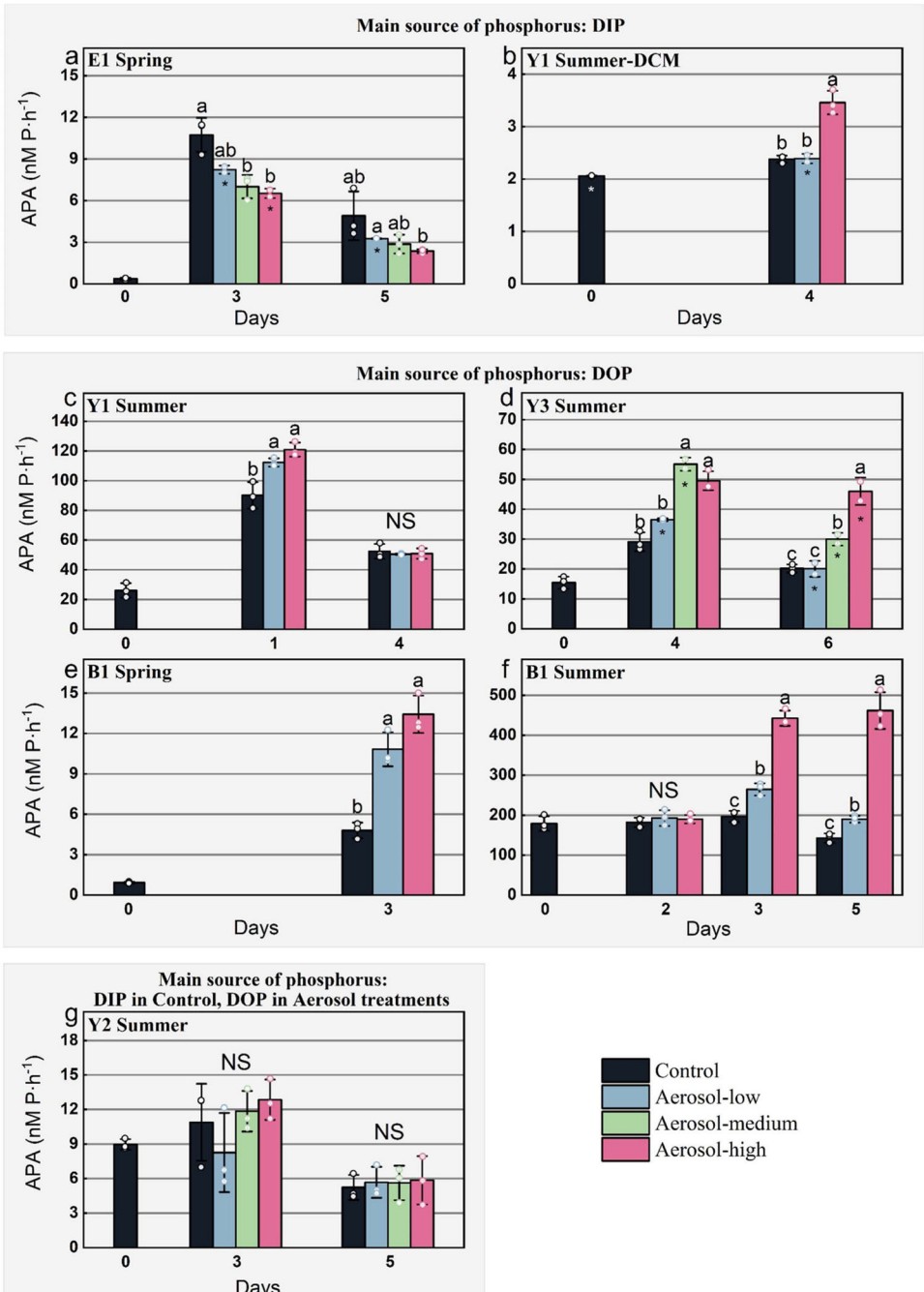

**Fig. 3 | The activity of alkaline phosphatase (APA) in response to aerosol addition.** APA in the experiments where dissolved inorganic phosphorus (DIP) was the main source of phosphorus (**a**, **b**), dissolved organic phosphorus (DOP) was the main source of phosphorus (**c**–**f**), and DIP was the main source of phosphorus in control and DOP was the main source of phosphorus in aerosol treatments (**g**). Between different groups at the same day of incubation, indistinguishable means labeled with the same letter ($p < 0.05$) and NS = not significant (one-way ANOVA and Turkey HSD post hoc test). Data are presented as mean values +/- SD, $n = 3$ samples for all except where indicated by an asterisk ($n = 2$ samples). Source data are provided as a Source Data file.

DIP deficiency or P* (DIP–DIN / 16) in terms of DOP consumption or accumulation[44,67] (Fig. S12), suggesting the important role of phytoplankton biomass (Chl *a*) in affecting DOP utilization in coastal seas. Moreover, APA always increased in our experiments when $Log_{10}$ (Chl *a* / DIP) was ≥ 1.20 (Fig. 6a). This suggests that $Log_{10}$ (Chl *a* / DIP) might serve as a new index which can be applied globally because our experiments covered surface marine ecosystems from oligotrophy to eutrophy (Figs. 6b and S2–3).

We also found that the relative change of APA after aerosol and riverine water additions was positively correlated with the aerosol and riverine water derived-increase of DIN:DIP (Fig. 5b). APA showed a similar positive response (differences no larger than ±21.0%) to DIN and aerosol additions with same DIN content at Y1, B1$_{Spr}$, and B1$_{Sum}$ (Fig. 3c, e, f and S5a-c). Although the role of essential trace metals such as Zn and Co for the APA was not tested in our study, the APA did not significantly respond to Fe or N+Fe additions (Figs. S4 and S5), and the APA in the aerosol treatment was comparable to those of the riverine water treatment (Fig. 3e-f and S6a-b), suggesting that other trace metals usually enriched in aerosol and riverine water were likely not having a substantial impact on our results. Indeed, the concentrations of trace

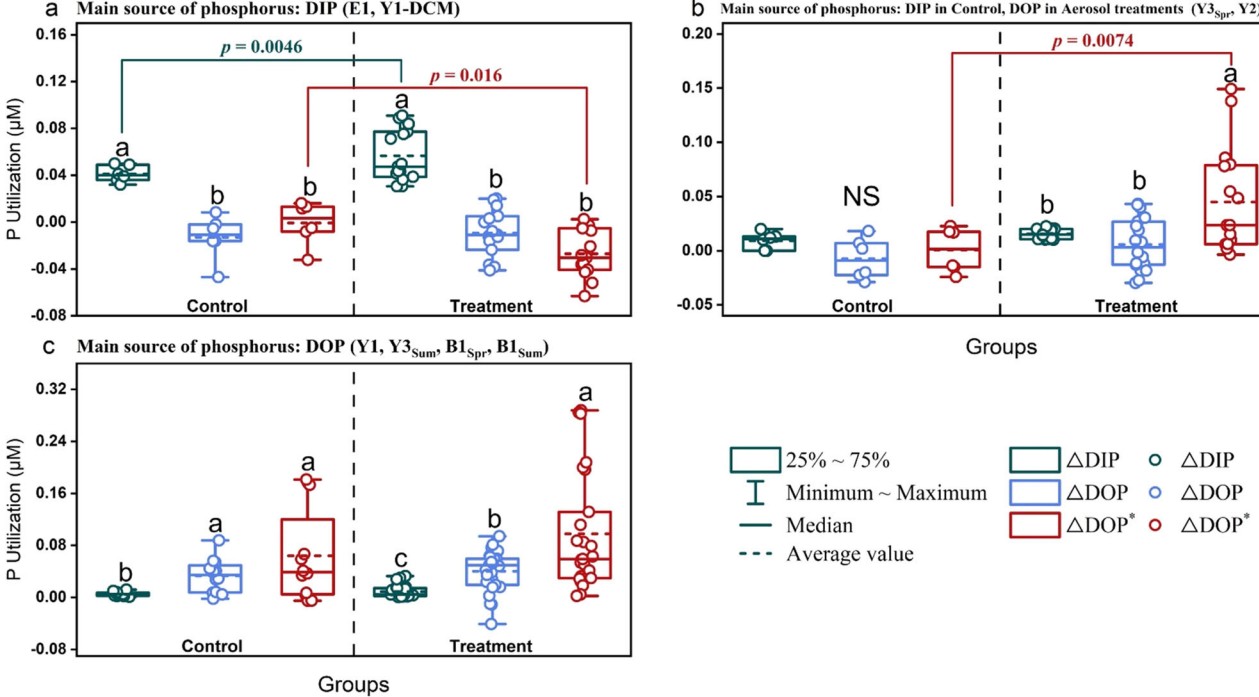

**Fig. 4 | Phosphorus utilization in response to aerosol additions.** phosphorus (P) utilization in the experiments where dissolved inorganic phosphorus (DIP) was the main source of P (**a**), DIP was the main source of P in control and dissolved organic phosphorus (DOP) was the main source of P in aerosol treatments (**b**), and DOP was the main source of P (**c**). DIP utilization ($\Delta$DIP) and net DOP utilization ($\Delta$DOP) were calculated during the incubations from the change of DIP and DOP concentrations, respectively. Potential maximum DOP utilization ($\Delta DOP^* = \Delta DIN/16 - \Delta DIP$) was calculated by the DIN ($\Delta$DIN) and DIP ($\Delta$DIP) utilization during the incubations using the Redfield ratio of 16:1 for N:P. Between different P utilization ($\Delta$DIP, $\Delta$DOP, $\Delta DOP^*$) in the same group, indistinguishable means labeled with the same letter and NS = not significant (one-way ANOVA and Turkey HSD post hoc test). Significant differences ($p < 0.05$) for P utilization between control and treatment groups are indicated with connecting colored lines and $p$ values (unpaired t-test, two-sided). $n$, number of samples. Source data are provided as a Source Data file.

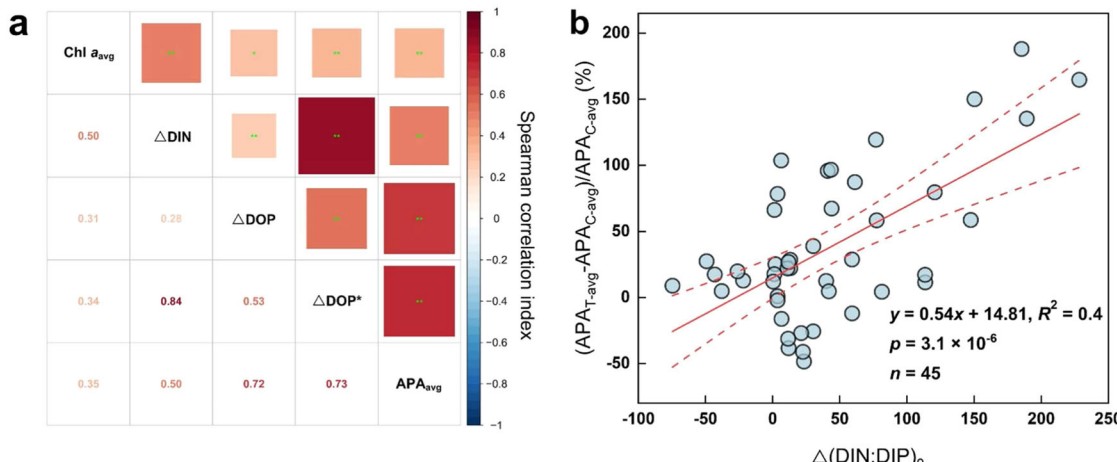

**Fig. 5 | Correlations to estimate how the anthropogenic nitrogen pump affects the utilization of dissolved organic phosphorus (DOP). a** Heatmap of Spearman correlations between measured variables, including the average Chlorophyll $a$ over the time during the experimental incubations (Chl $a_{avg}$), the utilization of dissolved inorganic nitrogen ($\Delta$DIN), the net utilization of DOP ($\Delta$DOP), the potential maximum utilization of DOP ($\Delta DOP^*$), and the average activity of alkaline phosphatase over a time course during the incubations (APA$_{avg}$); $n = 3$ samples. The blue and red squares represent positive and negative correlations, respectively. The numbers represent the Spearman correlation index. The * and ** indicate 5% ($p < 0.05$) and 1% ($p < 0.01$) significant level, respectively (Nonparametric Spearman correlation test (r), two-sided). No adjustments were made for multiple comparisons. **b** Correlation between treatment-induced change of average APA over time course during incubation ((APA$_{T-avg}$-APA$_{C-avg}$)/APA$_{C-avg}$) and change of DIN:DIP at the beginning of incubation ($\Delta$(N:P)$_0$). APA$_{C-avg}$ = APA in control group; APA$_{T-avg}$ = APA in treatment (aerosol and riverine water additions). The red solid line represents the best fit line of linear regression and the dashed lines represent the 95% confidence interval. $p$ values from a two-sided test are shown. $n$, number of samples. Source data are provided as a Source Data file.

metals are generally deemed to be elevated in coastal seas[30,68–70]. Our results were corroborated by a study in the Western English Channel where the increase in APA was associated with increased river runoff resulting in elevated N:P ratios[18]. Thus, apart from the atmospheric deposition, river runoff likely also enhances DOP utilization in the coastal seas through a similar mechanism. However, air pollutants relative to river runoff will have an impact over larger geographical areas because of atmospheric long-range transport[41].

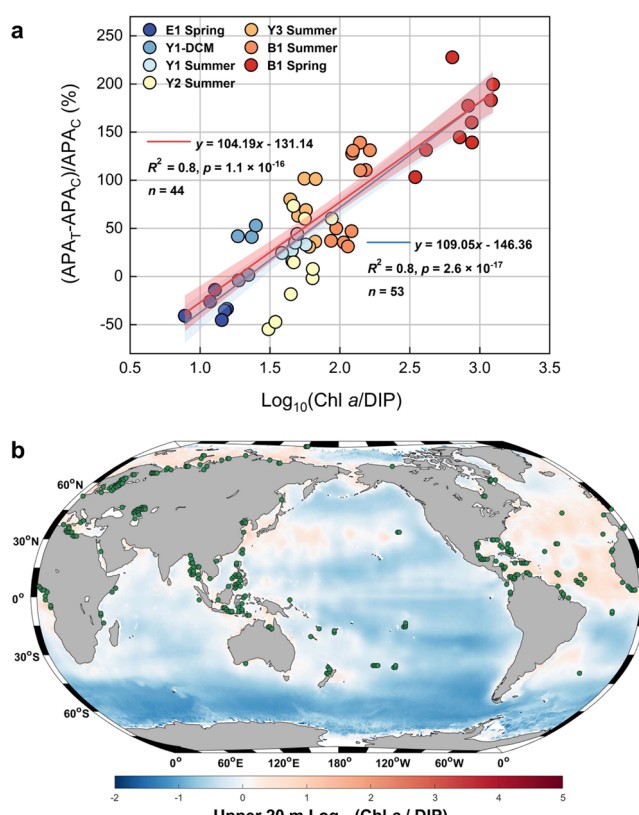

**Fig. 6 | The impact of atmospheric deposition and river runoff on the utilization of dissolved organic phosphorus (DOP) in coastal oceans on a global level.**
**a** Correlation between the relative change of activity of alkaline phosphatase (APA) and the ratio of chlorophyll $a$ to dissolved inorganic phosphorus [$Log_{10}$ (Chl $a$ / DIP)] for control (APA$_C$) and treatment groups (APA$_T$). The blue and red lines (best-fit line) with shades (95% confidence level) represent the results of the linear regression with and without data of the oligotrophic station (Y2), respectively. $p$ values from a two-sided test are shown. $n$, number of samples. **b** The distribution of $Log_{10}$ (Chl $a$ / DIP) in the global surface ocean (upper 20 m) calculated using average Chl $a$ data from SeaWIFS and MODIS-Aqua and average DIP data from WOA 2018. Green points represent $Log_{10}$ (Chl $a$ / DIP) larger than 1.20, which is considered the threshold for DOP utilization stimulated by atmospheric deposition. The color scale indicates the climatological $Log_{10}$ (Chl $a$ / DIP) in the global ocean (1° × 1°). Source data are provided as a Source Data file.

The seawater temperature, although an important factor determining enzyme activity[20], is likely not the main determining factor for DOP utilization although it has an influence due to the temperature-dependence of enzyme activity. For example, in the subtropical Atlantic at relatively high seawater temperatures (21–29 °C) and off the coast of northern Oregon at relatively low seawater temperatures (11–12 °C), AP was activated regardless of the water temperature when phytoplankton experienced P stress[13,17].

Collectively, our results demonstrate that aerosol and riverine water enhance the APA primarily by providing N, leading to an increase of DIN:DIP ratios and Chl $a$ concentrations. Thus, our results suggest that aerosol deposition and river runoff activate a nitrogen pump (Anthropogenic Nitrogen Pump, ANP) exacerbating DIP deficiency, which stimulate the utilization of DOP (Fig. 7). Moreover, using the new index [$Log_{10}$ (Chl $a$ / DIP)], we identified that the regions with positive $Log_{10}$ (Chl $a$ / DIP) values were generally corresponding to low DIP and DOP concentrations, indicating the geographic location where DIP deficiency and DOP utilization maybe observed[4,44]. Higher $Log_{10}$ (Chl $a$ / DIP) ($\geq$1.20) values were mainly identified in coastal seas at a global scale where the imbalance of N and P nutrients has become increasingly prevalent in recent years[39],

including the Mediterranean Sea, the Gulf of Mexico, and the Red Sea where APA has been detected[1] and a wide range of coastal seas where no relevant studies have been done yet (Fig. 6b). Therefore, the ANP has the potential to enhance DOP utilization across global coastal seas. The Arctic Coastal Seas appears to be a special case because although the Arctic shelf generally is characterized by elevated DIP and DOP concentrations[4,44], higher $Log_{10}$ (Chl $a$ / DIP) ($\geq$1.20) values were also found. This might be related to the decrease of sea-ice cover due to rapid global warming in the Arctic, which reduces nutrient concentrations in surface waters[71]. Furthermore, our study indicates the vital role of DOP as the main phosphorus source in enhancing DIN uptake by phytoplankton with consequences for the N:P ratios (Figs. 2, 5, and S13). Thus, the utilization of DOP can support phytoplankton to cope with phosphorus limitation[5,72], and can even shift the limiting nutrient to N as shown at station B1$_{Spr}$ in the central Bohai Sea (Fig. 2a and Table S2), which usually is characterized as an ecosystem that limits phytoplankton productivity by the availability of P[53].

Increasing anthropogenic N input in recent years at a global level has elevated the N:P ratios in coastal seawater, and progressively expanded beyond the coastal regions[23,24,39,43]. In the context of decreased prevalence of N deficiency over time, the impact of N input is gradually shifting from alleviating N limitation to increasing DIP limitation, especially in coastal regions[39,43,73]. Over the past four decades, the disproportionate and sustained input of N relative to P into the China Coastal Seas has led to the transition from N to P limitation in extensive coastal areas[43], which has also been observed around the globe wherever coastal seas receive significant input of N either by atmospheric deposition or through riverine input. If this N input continues under a 'business-as-usual' scenario, coastal biogeochemical processes require re-examination because our data suggest that DOP utilization by phytoplankton will become increasingly important with consequences for P-cycling in coastal seas. Atmospheric N input to the ocean has significantly increased from 1850 (10 Tg N yr$^{-1}$) to 2005 (39 Tg N yr$^{-1}$), and is expected to remain elevated until at least 2100 (-17%–+8%)[23] at a business-as-usual scenario. Thus, the Anthropogenic Nitrogen Pump is at least expected to have a continued impact on the biogeochemistry of coastal seas globally, and its role might even become more prominent under increasing N deposition[22] unless we reduce the input of terrigenous N-rich material such as part of aerosols and riverine water.

## Methods
### Collection and chemical analysis of atmospheric aerosols and riverine water
Atmospheric aerosols were collected at the Atmospheric Environment Monitoring Station in the Laoshan campus of the Ocean University of China, Qingdao, China (36.16° N, 120.50° E), about seven kilometers off the coast of the Yellow Sea (Fig. 1). The sampling was conducted under haze weather (visibility <10 km, relative humidity <90%) in January 2018 (Aerosol 1), January 2019 (Aerosol 2), and June 2019 (Aerosol 3) (Table S1). The aerosols were collected onto an acid washed filter (Whatman, 41#) through a high-volume sampler (KC-1000, Laoshan electronic instrument company of Qingdao) operated at a flow rate of 1.0 m³·min$^{-1}$ for about 20 hours uninterruptedly, and a blank was simultaneously collected at a flow rate of 0 m³·min$^{-1}$ under the same condition[74].

The aerosol sample was divided up into several aliquots for chemical analyses and the on-board microcosm experiments. For chemical analyses, two parts of the filter were extracted ultrasonically with deionized water (18.2 MΩ cm) in an ice-water bath for 40 minutes and the extracts was filtered instantly through a PES membrane (aperture 0.45 μm). The first extract was used for nutrient (NO$_3^-$ + NO$_2^-$, NH$_4^+$, DIP, and Si (OH)$_4$) analysis using an QuAAtro continuous-flow analyser (SEAL Analytical)[47]. The second extract was digested using 15.5 mol·L$^{-1}$

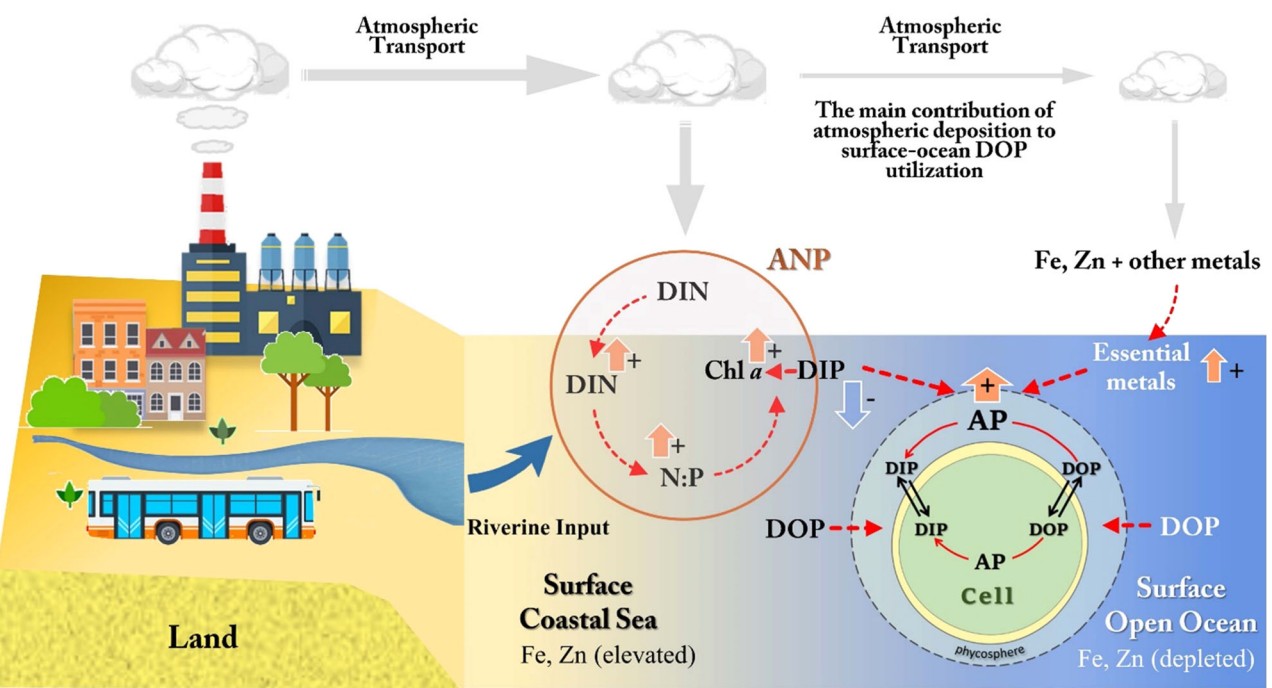

**Fig. 7 | Conceptual diagram of the anthropogenic nitrogen pump (ANP).** In coastal seas, atmospheric deposition and river runoff are mainly contributing dissolved inorganic nitrogen (DIN) whereas in the surface open ocean, atmospheric deposition is mainly contributing essential trace metals for enhancing the activity of the alkaline phosphatase (APA). DIN therefore increases in surface waters of the coastal seas leading to higher N:P ratios, enhanced phytoplankton growth, and the uptake of dissolved inorganic phosphorus (DIP). Once DIP is below a certain threshold, the APA significantly increases. This process leads to an increase in the utilization of dissolved organic phosphorus (DOP).

$HNO_3$ and then evaporated. The solution was used for dissolved trace metal analysis using an Agilent 7500c ICP-MS[75]. The detection limit was $0.4\ \mu g \cdot L^{-1}$ for Fe and was $0.09\ \mu g \cdot L^{-1}$ for other metals. The third part of the filter was used for aerosol amendments in the microcosm experiments. It was extracted before the microcosm experiments took place following the same procedure as described above for the nutrient analysis[76,77].

The riverine water was collected using a surface-water collector in the Xiaoqing River in 2019. The riverine water was filtered through acid-washed acetate fiber membranes (0.45 μm pore size) and stored in HDPE bottles at -20 °C. It was defrosted in the dark at room temperature (18–20 °C) before measuring the chemical composition. Nutrient concentrations ($NO_3^- + NO_2^-$, $NH_4^+$, DIP, and Si $(OH)_4$) of riverine water were then measured using the QuAAtro continuous-flow analyser (SEAL Analytical)[47].

### Set up of microcosm experiments
Eight on-board microcosm experiments were carried out at five stations in the East China Sea, the Yellow Sea, and the Bohai Sea using surface seawater collected in spring (March−April) and summer (July−August) during the cruises 2018 by R/V Dongfonghong 2 and 2019 by R/V Beidou, respectively (Fig. 1). Surface seawater (3–5 m depth) as well as water from the DCM layer of Y1 were collected by Teflon-coated Niskin bottles assembly on a SeaBird CTD and pre-filtered through 200 μm acid-washed nylon mesh to eliminate large zooplankton. They were transferred into transparent polycarbonate bottles, and instantly spiked with the extract of aerosol sample (20 L bottle experiments, Nalgene 2251-0050) and combinations of nutrients / trace metals ($KNO_3$, 2 μM; $KH_2PO_4$, 0.2 μM; $FeCl_3$, 2 nM) (2 L bottle experiments, Nalgene 2015-2000), respectively (Table S4). Aerosol 1 was used for the microcosm experiments at Y2, $Y3_{Spr}$, and $Y3_{Sum}$, Aerosol 2 for those at E1 and $B1_{Spr}$, and Aerosol 3 for those at Y1 and $B1_{Sum}$ (Table S4). Riverine water amendments were also set up in the 20 L bottle experiments at $B1_{Spr}$ and $B1_{Sum}$, to make a comparison with aerosol treatments under the condition of same DIN supply. The added amount of aerosol and riverine water was calculated based on their DIN ($NO_3^- + NO_2^- + NH_4^+$) concentrations (0.5, 1, 2 μM, Table S4). Three parallel experiments ($n = 3$ samples) were set up for each treatment. The bottles were placed in three microcosm tanks and incubated for 4–6 days for 20 L bottle experiments and 3–5 days for 2 L bottle experiments with flow-through surface seawater to keep the temperature stable and like in-situ conditions. The experiments were covered by photomask to attenuate photosynthetically active radiation by around 40%[78].

### Sampling and measurements
To mix the substances in bottles adequately, samples were taken after gentle shaking. Subsamples for the Chl *a* and nutrient analysis were taken every 24 h during most of the incubations. Samples for APA analysis were mostly taken at the beginning, middle and the end of the incubations, and samples for the DOP measurements were taken at the beginning and end of the incubations.

### Chlorophyll *a* measurements
Triplicate samples of 200–300 mL were filtered through 20 (nylon, Millipore), 2 (polycarbonate, Whatman) and 0.2 (polycarbonate, Whatman) μm aperture sized membranes in time series, and total Chl *a* concentrations were calculated by combining size-separated Chl *a* concentrations for the 20 L incubation systems, and 100–150 mL were filtered through a GF/F membrane (Whatman) to measure total Chl *a* for the 2 L incubation systems. The pigments on the membranes were extracted with 90% acetone at -20 °C for 20 h in the dark. The extract was used for measuring Chl *a* using a 7200-000 Trilogy fluorometer (Turner Designs)[47].

### Inorganic and organic nutrients
100 mL samples of 20 L bottle experiments were filtered through 0.45 μm acid-washed acetate membrane, and the filtrate was stored in acid-

washed HDPE bottles at -20 °C. The samples were melted at room temperature 48 hours before measuring them. 50 mL of filtrate was used for determining inorganic nutrients ($NO_3^-$ + $NO_2^-$, $NH_4^+$, DIP, $Si(OH)_4$) using a QuAAtro continuous-flow analyser (SEAL Analytical)[47]. The detection limits of $NO_3^-$ + $NO_2^-$, $NH_4^+$, and $Si(OH)_4$ were ≤0.02 μM, ≤0.02 μM, and ≤0.03 μM, respectively[47]. The long optical fiber technology was used to measure DIP concentration and its detection limit was estimated to be ≤0.005 μM[79]. The error margin for the DIP analysis was 1.39% of the average value of 0.06 μM, which represents the relative standard deviation of five replicates. To measure total dissolved phosphorus (TDP) and dissolved organic phosphorus (DOP), 40 mL filtrate and 4 mL oxidizing agent were put into a Teflon digestion tank and were disintegrated at 120 °C for 20 min in an autoclave to enable the detection of TDP. The oxidizing agent was made by dissolving 10 g $K_2S_2O_8$ and 6 g $H_3BO_3$ in 200 mL prepared NaOH solution (0.375 mol·$L^{-1}$)[80]. The detection limit of TDP was 0.02 μM, and the error margins for TDP analysis was 0.77% of the average value of 0.20 μM through the same calculating method as that of DIP. DOP was estimated by subtracting DIP from TDP. The detection limit (0.025 μM) for DOP analysis was calculated by the sum of those of DIP and TDP ($DL_{DIP}$ + $DL_{TDP}$). The error margin for DOP analysis (1.27% of the average value of 0.14 μM) was calculated using the standard deviations (SD) and average values (avg) of five replicates for TDP and DIP analysis (Eq. 4).

## Activity of Alkaline Phosphatase (APA)

4-methylumpholone phosphate (MUF-P) was used as the substrate to analyze APA. MUF-P was converted to $PO_4^{3-}$ and 4-methylumpholone (MUF) by the alkaline phosphatase, in which MUF showed fluorescence when pH is above 10[81]. For 2 L bottle experiments in 2019 and 20 L bottle experiments in 2018 in summer and in 2019, 45 mL sample with 200–1000 nM MUF-P was incubated in the microcosm tank in the dark after gently mixing[11]. The fluorescence and therefore the APA (nM P·$h^{-1}$) were calculated estimated by time-course measurements (0, 1.5, and 3 h) using the 7200-000 Trilogy fluorometer (Turner Designs) with the long wavelength UV module (P/N 7000-967) after adding borate buffer (pH = 10.8–11.2) to the samples[82]. The concentration of MUF corresponding to the fluorescence was calculated using the standard curve (0–1 μM) configured with MUF-2Na[5,82].

## Data analyses

As the seawater was collected at different nutrient status and phytoplankton growth stage, the Chl $a$ concentration and its response to different treatment were variable (Figs. S1 and S2). The response of phytoplankton to nutrient, aerosol, and riverine water additions was there quantified by $\Delta R$[61], which was expressed as:

$$\Delta R = Log_{10}(Chl\, a_{t-avg}/Chl\, a_{c-avg})/t \quad (1)$$

where Chl $a_{c-avg}$ and Chl $a_{t-avg}$ stand for the average Chl $a$ concentration in the control and N, P, N + P, aerosol, or riverine water treatment groups, respectively, over the time course during the microcosm experiments ($n = 3$ samples); $t$ represents duration of microcosm experiment in days.

Nutrient utilizations were estimated by the decrease of nutrient concentrations (DIN, DIP, or DOP) over time during the incubations:

$$\Delta X = X_0 - X_t \quad (2)$$

where the $X_0$ and $X_t$ represent the nutrient concentrations at the beginning and the end of the incubations, respectively ($n = 3$ samples) (Figs. S3 and S8).

$P^* = DIP - DIN / 16$ is often used to determine excess of available dissolved inorganic phosphorus (DIP). A positive correlation between

surface DOP concentration and $P^*$ was found on a global level[11,44]. The potential maximum DOP utilization during the incubations was therefore calculated according to:

$$\Delta DOP^* = \Delta DIN/16 - \Delta DIP \quad (3)$$

where the $\Delta DIN$ and $\Delta DIP$ represent the change of DIN and DIP concentrations, respectively, during the incubations. The number 16 represents the standard Redfield Ratio for N:P requirement of phytoplankton. Although the nutrient uptake ratio can vary[83], the POC:PON ratios in this study (4.07–10.50, mean = 6.96) were close to Redfield ratio (C:N = 6.63) and positive correlations were measured between $\Delta DOP^*$, $\Delta DOP$, and APA, indicating the robustness of the parameter $\Delta DOP^*$ in this study (Fig. 5a and S14).

The error margin for DOP analysis was calculated using the standard deviations (SD) and average values (avg) of five replicates for TDP and DIP analysis according to:

$$\frac{\sqrt[2]{SD_{TDP}^2 + SD_{DIP}^2}}{TDP_{avg} - DIP_{avg}} \times 100\% \quad (4)$$

## Statistical analyses

The response of Chl $a$ to nutrient additions (Fig. 2a), the response of APA to nutrient, aerosol, and riverine water additions (Figs. 3 and S4–7), and the differences in P utilization (Fig. 4, S6, and S9) were evaluated using one-way ANOVA. A Tukey HSD post hoc test was used to identify statistically significant differences. We used the "aov" function for one-way ANOVA, and the "TukeyHSD" function for Tukey HSD post hoc test in the R package "stats"[61]. Unpaired t-test (two-sided) was used to identify statistically significant differences between the response of Chl $a$ to treatments (aerosol and riverine water) in spring and summer (Fig. 2b), and between the P utilization in control and treatments (aerosol and riverine water) (Fig. 4). We used the "t.test" function for unpaired t-test in the R package "stats"[84]. Nonparametric Spearman correlation test (r) (two-sided) was used to evaluate the correlation between parameters (Chl $a$, $\Delta DIN$, $\Delta DIP$, $\Delta DOP$, and $\Delta DOP^*$) that could impact DOP utilization (Fig. 5a). Linear regression was used in Figs. 5b, 6a, and S12. To explore the statistically significant threshold of APA and DIP concentrations, a segmented regression was conducted through R package "segmented"[85]. To investigate $Log_{10}$ (Chl $a$ / DIP) in the global surface ocean (upper 20 m, 1° × 1°), we obtained Chl $a$ concentration data at 9-km spatial resolution from SeaWiFS[86] and MODIS-Aqua[87] ocean color observations and the climatological mean for DIP data from World Ocean Atlas 2018[88].

## Reporting summary

Further information on research design is available in the Nature Portfolio Reporting Summary linked to this article.

## Data availability

Data of microcosm experiments, chemical analyses of aerosols and riverine water used in this study are available in figshare under accession code [https://figshare.com/s/aed72ac92028a0e0f705]. The Level-3 satellite data (Chlorophyll $a$) used in this study are available in the SeaWIFS database [https://oceancolor.gsfc.nasa.gov/data/seawifs/] and MODIS-Aqua database [https://oceancolor.gsfc.nasa.gov/data/aqua/]. The data used for generating the climatological mean for the dissolved inorganic phosphorus are available in the World Ocean Atlas 2018 database [https://www.ncei.noaa.gov/products/world-ocean-atlas]. Maps were generated in matlab using M_Map (https://www.eoas.ubc.ca/~rich/mapug.html). Source data are provided with this paper.

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

## Acknowledgements

This work was funded by NSFC-Shandong Joint Fund (U1906215), National Key R&D Program of China (2022YFF0803000), National Natural Science Foundation of China (NSFC) (41906119, 41876125, 42176022), Fundamental Research Funds for the Central Universities (202072001). Data and samples for microcosm experiments were collected onboard of R/V "Dongfanghong 2", R/V "Beidou", and R/V "Lanhai101" implementing the open research cruise NORC2018-1, NORC2019-1, NORC2020-1, and NORC2023-1 supported by NSFC Shiptime Sharing Project (project number: 41576118, 41849901, 41949901, and 42249901). T.M. acknowledges funding provided by NERC under grant NE/W005654/1 and the School of Environmental Sciences for partial support to supervise Haoyu Jin and to write and edit this paper.

## Author contributions

H.J., C.Z. and H.G. conceptualized the work and designed the study. H.J. drafted the first version of the manuscript. C.Z., T.M., X.D., Y.Z., L.M., X.Y., Y.G. and H.G. contributed to later visions of the manuscript. H.J., C.Z., Q.W., S.M. and X.D. performed the field investigation. F. S. contributed additional experiments and analyses to address reviewers' comments. H.J., C.Z., S.M. and H.G. conducted the data analysis and visualization. C.Z. and H.G. received funding to conduct this work. C.Z., T.M. and H.G. were responsible for supervision of H.J. All authors gave feedback on the manuscript and have approved the final version of the paper.

## Competing interests

The authors declare no competing interests.
