## [Peer Review File · Nature Communications]

Atmospheric deposition and river runoff stimulate the utilization of dissolved organic phosphorus in coastal seasREVIEWER COMMENTS

Reviewer #1 (Remarks to the Author):

REVIEW of a manuscript entitled "Atmospheric aerosols stimulate the utilization of dissolved organic phosphorus in coastal seas" by Haoyu Jin et al.

Jin and his colleagues conducted a study that demonstrated how aerosol additions to the coastal waters near China enhance the utilization of DOP by phytoplankton. This finding is relevant due to the substantial increase in anthropogenic N input to this region of the Pacific. While one would expect an increase in phytoplankton biomass in response to higher anthropogenic N input, the exact mechanism for this biomass increase has not been fully addressed. Specifically, the hypothesis that the addition of N input maximizes the utilization of the DOP pool to increase biomass has not been experimentally proven until now. The paper aimed to investigate this outstanding question and, to some extent, the authors achieved their goal. I am hopeful that the authors' efforts will yield further progress.

In preparing the revised manuscript, I have several reservations that the authors should consider.

(1) One crucial aspect to address in this study is the shift in the nutrient regime shift in the marginal seas of the northwestern Pacific, particularly the transition from N limitation to P limitation, which can be attributed to the addition of anthropogenic N input. A recent L&O paper by Moon et al. (66, 2021, 914-924) provides valuable insights in this regard. The authors observed a change in nutrient dynamics during the 1990s, where the prevalence of N deficiency decreased over time. Concurrently, they found the emergence of P deficiency in the vicinity of the Changjiang estuary, which progressively expanded beyond the estuary. Given the significance of the P-limitation condition in this work, including a paragraph outlining this nutrient regime shift would not only justify your work but also emphasize the importance of your finding.

(2) I challenge use of $\log(\text{Chl } a/\text{DIP})$: The use of $\log(\text{Chl } a/\text{DIP}) = 1.2$ appears to be subjectively chosen. To me this ratio simply indicates the conditions of low DIP concentration and high chl a concentration. I suggest using $N^* = [N] - 16 [P]$ as an alternative approach. N^* is direct measure of either N excess (P deficit) or N deficit (P excess). Can you recalculate all data used in Figure 5a? N and P concentration data in most locations are likely available. I like to see how the results based on N^* differ from those based on $\log(\text{Chl } a/\text{DIP}) = 1.2$.

(3) Addition of river water (rich in N relative to P) to seawater samples can be problematic in microcosm experiments: The introduction of river water into the seawater samples resulted in a decrease in salinity. This alteration in salinity subsequently impacted the growth of phytoplankton. How did you take this fact into consideration when you interpret the results?

(4) ΔDOP^* is not entirely clear to me relative to ΔDOP : Please provide more detailed explanation

in comparison with delta DOP.

(5) Figures 2 and 3 are very hard to follow: I am extremely unhappy with Figures 2 and 3, as they lack intuitive appeal. Considering that these figures encompass the key findings of this work, I strongly encourage the authors to invest considerable effort in revising them. Upon reading the figure legends, I encountered considerable difficulty in interpreting the figures. Ideally, the figures should be self-explanatory by visual examination alone.

Reviewer #2 (Remarks to the Author):

This is an interesting paper which addresses growing interest in the cycling of nutrients in marine systems. The data are robust and will add to knowledge of the biogeochemical processes involving P and interrogate the relationship between DIN and DOP and the role of N. My reservations on recommending the paper for publication are mainly how much this comprehensive dataset advances knowledge as the idea that low DIP or an increased N:P ratio will induce APA and utilisation of DOP is well established and accepted, though as I have noted there are also reports of constitutional uptake of DOP (see separate table of comments). Rees et al (2009) doi:10.1016/j.ecss.2008.12.005 observed an increase in APA in the Western English Channel and this corresponded with a salinity decrease related to increased riverine inputs, increasing the N:P ratio in the sampling area. If these inputs are riverine or atmospheric it stands to reason that DOP utilisation could be stimulated. The last sentence on curbing nitrogen inputs is vague and could be constrained using estimates of anthropogenic inputs, such as Jickells et al (2017) doi:10.1002/2016GB005586. This could strengthen Figure 5, where N deposition is key.

Reviewer #3 (Remarks to the Author):

Jin et al present the results from shipboard nutrient, aerosol, and river water amendment microcosm experiments to investigate the controls on alkaline phosphatase activity (APA) and DOP utilization by phytoplankton in the coastal waters off the East Asian continent. The title and abstract focus on the role of aerosols and the macro and micro nutrients they contain in stimulating APA and DOP utilization in the microcosms. The authors conclude that “atmospheric aerosols stimulate the utilization of dissolved organic phosphorus in coastal seas” and that this arises primarily by depleting ambient DIP when stoichiometric larger additions of DIN are added by aerosols, with phytoplankton turning to DOP to satisfy their P demand.

In my opinion, the authors fail to convincingly demonstrate 1) that DOP utilization is stimulated by aerosol addition, 2) that APA is stimulated by aerosol addition which would be required for DOP utilization, and 3) that the driving mechanism leading to enhanced DOP utilization by phytoplankton is the alteration of the ambient DIN:DIP by aerosol addition.

The study concludes that DOP utilization is stimulated by aerosol addition. The data in support of this is presented in Figure 3. Of the 8 stations presented for aerosol addition microcosms, only two conclusively demonstrate a DOP utilization effect that is greater than nil within the presented error bars (stations Y3 and B1). River water amendment microcosms were conducted at B1 with the magnitude of the DOP utilization effect comparable to that from aerosol addition, weakening the authors conclusion that aerosols alone are important for stimulating DOP utilization in coastal seas. As acknowledged by the authors, due to the weak support from interpreting enhanced DOP utilization in the microcosms by differences in measured DOP concentrations, the authors present a derived metric, DOP*, to support the notion that DOP utilization is enhanced by aerosol addition. DOP* represents the theoretical potential maximum in DOP utilization that might have occurred in the microcosm after computing the residual between the expected phytoplankton P demand (computed from the observed DIN utilization and Redfield N:P stoichiometry) and the observed DIP utilization. Using DOP* as the metric to gauge stimulation of DOP utilization by aerosol addition, raises the number of stations exhibiting a positive effect from 2 out of 8 to 4 out of 8, still not a very conclusive result in my opinion. The APA data presented in Figure 3 generally support the interpretations from using delta DOP or delta DOP*; i.e. in microcosms where a stimulation effect was observed for aerosol addition on DOP utilization, APA is higher and vice versa.

As for the conclusion that elevated DIN:DIP is responsible for elevated DOP utilization, the data in Figure 4b largely supports this by identifying a significant positive correlation between the relative APA stimulation (expressed as a percentage relative to the controls) versus DIN:DIP as the predictor variable. However, dissolved iron has also been identified as an important co-control on DOP utilization in marine surface waters (see Browning et al. 2017 Nat. Comm., Liang et al., 2022 Nat. Geo.). The authors need to demonstrate data and/or other arguments to include or rule out the potential control of enhanced DOP utilization by Fe addition from aerosols and possibly river water.

The authors should report the limit of detection and analytical uncertainty for DOP determinations after considering the additive uncertainties from the separate determinations of TDP and DIP. The precision on TDP analyses is unreported so it is impossible as a reviewer to determine if the delta DOP values reported in Figure 3 on the order of -30 to +90 nM are even larger than the analytical uncertainty on DOP determinations.

Lastly, I am excited by the identified $\log_{10}(\text{chl}/\text{DIP})$ metric for predicting APA and potentially DOP utilization (Fig. 5). I believe many marine biogeochemists will find its predictions at the global scale to be useful in helping predict the role of DOP in the local to regional biogeochemistry across the global ocean.

Response to reviewers' comments. Text in *italics* (copied from revised manuscript) represents how we have addressed the reviewers' suggestions in our revised manuscript. Our response in blue.

Reviewer #1:

1. One crucial aspect to address in this study is the shift in the nutrient regime shift in the marginal seas of the northwestern Pacific, particularly the transition from N limitation to P limitation, which can be attributed to the addition of anthropogenic N input. A recent L&O paper by Moon et al. (66, 2021, 914-924) provides valuable insights in this regard. The authors observed a change in nutrient dynamics during the 1990s, where the prevalence of N deficiency decreased over time. Concurrently, they found the emergency of P deficiency in the vicinity of the Changjiang estuary, which progressively expanded beyond the estuary. Given the significance of the P-limitation condition in this work, including a paragraph outlining this nutrient regime shift would not only justify your work but also emphasize the importance of your finding.

Response:

Thanks for your helpful suggestion. We have implemented your idea as follows (copied from the revised version of our manuscript):

“Hence, coastal seas are likely more affected than the open ocean in terms of an increased input of N, which can lead to elevated N:P ratios^{39,40} and therefore potentially a gradual shift from N deficiency to P deficiency⁴¹.”

“In contrast to the increased atmospheric N deposition, DIP concentrations have been continuously decreasing since the mid-1990s in the Yellow Sea, hence causing P deficiency^{41,56}”

“Increasing atmospheric and riverine N input in recent years at a global level has elevated the N:P ratios in coastal seawater, and progressively expanded beyond the coastal regions^{24,35,40,41}. In the context of decreased prevalence of N deficiency over time, the impact of N deposition is gradually shifting from alleviating N limitation to increasing DIP limitation, especially in coastal regions^{35,41,71}. Over the past four decades, the disproportionate and sustained input of N relative to P into the China Coastal Seas has led to the transition from N limitation to P limitation in extensive areas⁴¹.”

2. I challenge use of log (Chl a/DIP): The use of $\log(\text{Chl a/DIP}) = 1.2$ appears to be subjectively chosen. To me this ratio simply indicates the conditions of low DIP concentration and high chl a concentration. I suggest using $N^* = [N] - 16 [P]$ as an alternative approach. N^* is direct measure of either N excess (P deficit) or N deficit (P excess). Can you recalculate all data used in Figure 5a? N

and P concentration data in most locations are likely available. I like to see how the results based on N^* differ from those based on $\log(\text{Chl } a/\text{DIP}) = 1.2$.

Response:

We implemented the idea to calculate N^* but found no significant correlation between the relative change of APA and DIP deficiency (Fig. R1), even if we excluded site E1, which was sampled at the end of the spring bloom. Although N^* has been widely used in various studies (Browning et al., 2017; Kim et al., 2014; Liang et al., 2022), as we mentioned in this study, Chl *a* is equally important in terms of affecting APA because it is an indicator of phytoplankton biomass. The same N^* is likely corresponding to different Chl *a* concentrations likely resulting in different levels of P stress and thus varying APAs. For instance, when high N^* is accompanied by low Chl *a*, the small-sized phytoplankton are less affected by P stress (Marañón, 2015), as the residual P nutrient is likely matching their low P requirements. Nevertheless, to address your suggestion in our manuscript, a new figure has been added to the supplementary information (Fig. S10). This figure has also been discussed in the main manuscript as follows:

“In addition, the relative change of APA was not significantly correlated with N^ ($\text{DIN}/16\text{-DIP}$)³⁷ (Fig. S10), suggesting the important role of phytoplankton (Chl *a*) in affecting DOP utilization in coastal seas.”*

Fig. S10. Dissolved organic phosphorus utilization indicated by the activity of alkaline phosphatase (APA) and its correlation with phosphorus deficiency relative to nitrogen (N^{*}). Correlation between the relative change of APA after treatment and N^{*} (DIN/16-DIP), of which APA_C and APA_T represent APA on the day they began to enhance (mainly in the middle day of incubations) in the control and aerosol/riverine water treatments, respectively. DIN and DIP represent the concentration of dissolved inorganic nitrogen and dissolved inorganic phosphorus, respectively. 16 represent the Redfield ratio of N:P. The blue and red lines (best-fit line) with shades (95% confidence level) represent the results of the linear regression with and without data of E1 (covered by grey shades), respectively.

Browning, T. J., Achterberg, E. P., Yong, J. C., Rapp, I., Utermann, C., Engel, A., & Moore, C. M. (2017). Iron limitation of microbial phosphorus acquisition in the tropical North Atlantic, *Nat. Commun.*, 8, 15465.

Kim, I. N., Lee, K., Gruber, N., Karl, D. M., Bullister, J. L., Yang, S., & Kim, T. W. (2014). Increasing anthropogenic nitrogen in the North Pacific Ocean. *Science*, 346(6213), 1102-1106.

Liang, Z., Letscher, R. T., & Knapp, A. N. (2022). Dissolved organic phosphorus concentrations in the surface ocean controlled by both phosphate and iron stress. *Nature Geoscience*, 15(8), 651-657.

Marañón, E. (2015). Cell size as a key determinant of phytoplankton metabolism and community structure. *Annual review of marine science*, 7, 241-264.

3. Addition of river water (rich in N relative to P) to seawater samples can be problematic in microcosm experiments: The introduction of river water into the seawater samples resulted in a decrease in salinity. This alteration in salinity subsequently impacted the growth of phytoplankton. How did you take this fact into consideration when you interpret the results?

Response:

Because of your comment, we have estimated the impact of adding riverine water on the salinity in our experiments as good as we could. According to the result reported by Qing et al. (2013), the salinity of the surface seawater in the central Bohai Sea usually is between 30 - 33 Practical Salinity Units (PSU) across all sampled seasons. In our study, 30.25 mL and 121 mL of riverine water were added to River-low and River-high groups, respectively. The salinity of riverine water used in the microcosm experiments was 4.38 PSU. Therefore, the salinity of our 20 L incubation systems decreased by about 0.05 PSU for River-low group and by about 0.2 PSU for River-high group, which likely has a very weak impact on phytoplankton growth (Pilkaitytė et al., 2004) and APA (Fu and Bell, 2003).

Nevertheless, we have addressed this in the supplementary methods as follows:

“In addition, the salinity of the 20 L incubation systems decreased by about 0.05 PSU for the River-low group and 0.2 PSU for the River-high group, which likely had negligible impacts on phytoplankton growth.”

Qing, S., Zhang, J., Cui, T., & Bao, Y. (2013). Retrieval of sea surface salinity with MERIS and MODIS data in the Bohai Sea. *Remote Sensing of Environment*, 136, 117-125.

Pilkaitytė, R., Schoor, A., & Schubert, H. (2004). Response of phytoplankton communities to salinity changes—a mesocosm approach. *Hydrobiologia*, 513, 27-38.

Fu, F. X., & Bell, P. R. F. (2003). Effect of salinity on growth, pigmentation, N₂ fixation and alkaline phosphatase activity of cultured *Trichodesmium* sp. *Marine Ecology Progress Series*, 257, 69-76.

4. Delta DOP* is not entirely clear to me relative to delta DOP: Please provide more detailed explanation in comparison with delta DOP.

Response:

We have split the original Fig. 3 into two figures (Fig. 3 for APAs, Fig. 4 for P utilization) and updated them to make the parameters of P utilization much clearer. We have also tried to improve the explanation of ΔDOP^* in the manuscript and in the figure legend of Fig. 4 as follows:

“Note that ΔDOP as net DOP utilization did not show a significant response either to aerosol or

riverine water additions in most experiments (Figs. 4, S5c-d, and S8), which was likely ascribed to the high turnover rate of easily degradable DOP in coastal seas^{60,61}. We therefore defined ΔDOP^* ($\Delta DIN/16 - \Delta DIP$, Eq. 4) on the basis of DIN (ΔDIN) and DIP (ΔDIP) utilization (Eq. 3), and the Redfield ratio for N:P requirement (16) of phytoplankton.”

“Fig. 4 | Phosphorus utilization in response to aerosol additions. DIP utilization (ΔDIP) and net DOP utilization (ΔDOP) were calculated during the incubations from the change of DIP and DOP concentrations, respectively. Potential maximum DOP utilization ($\Delta DOP^* = \Delta DIN/16 - \Delta DIP$) was calculated by the DIN (ΔDIN) and DIP (ΔDIP) utilization during the incubations using the Redfield ratio of 16:1 for N:P. Significant differences ($p < 0.05$) between different P utilization (ΔDIP , ΔDOP , ΔDOP^*) in the same group are indicated by different letters and NS = not significant, and significant differences ($p < 0.05$) for P utilization between control and treatment groups are linked by coloured lines and indicated by coloured stars (one-way ANOVA and Turkey HSD post hoc test).”

5. Figures 2 and 3 are very hard to follow: I am extremely unhappy with Figures 2 and 3, as they lack intuitive appeal. Considering that these figures encompass the key findings of this work, I strongly encourage the authors to invest considerable effort in revising them. Upon reading the figure legends, I encountered considerable difficulty in interpreting the figures. Ideally, the figures should be self-explanatory by visual examination alone.

Response:

We agree and therefore have completely revised Figures 2 and 3 (see below). To make Fig. 2 clearer, we have moved N+Fe and N+P+Fe and the riverine water experimental groups to the supplement because they were just used for determining the nutrients that were mainly limiting the phytoplankton biomass. Furthermore, we have split Fig. 3 into two figures (Fig. 3 for APAs, Fig. 4 for P utilization) and re-designed them to make them clearer. We hope this has increased their visual appeal. We have also added a new figure to the supplement for illustrating P utilization at different stations (Fig. S8) and we moved the riverine data to the supplement as well (Fig. S5). Please see below for our new figures.

Fig. 2 | Response of Chl *a* concentration to nutrient (a) and aerosol (b) additions expressed by ΔR . ΔR ($\log_{10}(\text{Chl } a_{t\text{-avg}}/\text{Chl } a_{c\text{-avg}})/t$) is the \log_{10} ratio of the average Chl *a* concentration during the incubations in treatments relative to control groups. **a, The boxes with same colour used for data indicate the limiting nutrients (blue = N limitation, yellow = P limitation, red = N+P colimitation, and grey = not significant) ($p < 0.05$). Significant level was calculated using one-way ANOVA and Turkey HSD post hoc test.**

Fig. 3 | The activity of alkaline phosphatase (APA) in response to aerosol addition. Significant differences between different groups at the same day of the incubations are indicated by different letters and NS = not significant (one-way ANOVA and Turkey HSD post hoc test).

Error bars indicate standard deviations; $N=3$.

Fig. 4 | Phosphorus utilization in response to aerosol additions. DIP utilization (ΔDIP) and net DOP utilization (ΔDOP) were calculated during the incubations from the change of DIP and DOP concentrations, respectively. Potential maximum DOP utilization ($\Delta\text{DOP}^* = \Delta\text{DIN}/16 - \Delta\text{DIP}$) was calculated by the DIN (ΔDIN) and DIP (ΔDIP) utilization during the incubations using the Redfield ratio of 16:1 for N:P. Significant differences ($p < 0.05$) between different P utilization (ΔDIP , ΔDOP , ΔDOP^*) in the same group are indicated by different letters and NS = not significant, and significant differences ($p < 0.05$) for P utilization between control and treatment groups are linked by coloured lines and indicated by coloured stars (one-

way ANOVA and Turkey HSD post hoc test).

Fig. 5 Correlations to estimate how the atmospheric nitrogen pump affects DOP utilization. a, Correlations between measured variables (average Chl a over the time during the experimental incubations ($Chl\ a_{avg}$), ΔDIN , ΔDOP , ΔDOP^* , and the average activity of alkaline phosphatase over a time course during the incubations (APA_{avg}); $N = 3$). The blue and red squares represent positive and negative correlations, respectively. The numbers represent the Spearman correlation index. The * and ** indicate 5% ($p < 0.05$) and 1% ($p < 0.01$) significant level, respectively. b, Correlation between treatment-induced change of average APA over time course during incubation ($(APA_{T-avg} - APA_{C-avg}) / APA_{C-avg}$) and change of DIN:DIP at the beginning of incubation ($\Delta(N:P)_0$). APA_{C-avg} = APA in control group; APA_{T-avg} = APA in treatment (aerosol and riverine water additions). The red solid line represents the best fit and the dashed lines represent the 95% confidence interval.

Fig. S8 | Phosphorus utilization in response to aerosol additions. DIP utilization (ΔDIP), net DOP utilization (ΔDOP), and potential maximum DOP utilization ($\Delta DOP^* = \Delta DIN / 16 - \Delta DIP$). For DOP utilization, ΔDOP is calculated from the change of DOP concentration during the incubations, ΔDOP^* is calculated by the DIN (ΔDIN) and DIP (ΔDIP) utilization during the incubations, and 16:1 is the Redfield ratio of N:P. Significant differences ($p < 0.05$) between different P utilization (ΔDIP , ΔDOP , ΔDOP^*) in the same group are indicated by different

letters, and significant differences ($p < 0.05$) for P utilization between treatment and control groups are linked by coloured lines and indicated by coloured stars (one-way ANOVA and Turkey HSD post hoc test). Error bars indicate standard deviations; $N=3$.

Reviewer #2:

General comments:

1. My reservations on recommending the paper for publication are mainly how much this comprehensive dataset advances knowledge as the idea that low DIP or an increased N:P ratio will induce APA and utilisation of DOP is well established and accepted, though as I have noted there are also reports of constitutional uptake of DOP.

Response:

We agree that low DIP or an increased N:P ratio will induce APA and the utilisation of DOP, and this is very established knowledge indeed and it was also confirmed by our work, which is reassuring. However, the **novelty** of our work is to show the significance of atmospheric deposition in driving this biogeochemical process in coastal seas based on a case study in the China Coastal Seas, which we extended to the global level. This has not been shown before as most previous studies have focussed on open oligotrophic oceans when it comes to the role of atmospheric deposition in driving biogeochemical cycles.

To make this clearer, we have amended our text as follows:

“Over the past four decades, the disproportionate and sustained input of N relative to P into the China Coastal Seas has led to the transition from N to P limitation in extensive coastal areas⁴¹, which has also been observed around the globe wherever coastal seas receive significant input of dissolved inorganic nitrogen either by rivers or through atmospheric deposition. If this N deposition continues under a ‘business-as-usual’ scenario, coastal biogeochemical processes require re-examination because our data suggest that DOP utilization by phytoplankton will become increasingly important with consequences for P-cycling in coastal seas.”

For constitutional uptake of DOP, please see our response to your comment 2.0 down below.

2. Lines 52–54, There is also reported constitutional uptake of DOP (e.g. refs in Fitzsimons et al 2020. DOI: 10.1016/j.jembe.2020.151434 ; Mahaffey et al. 2014. doi: 10.3389/fmars.2014.00073

Response:

The term ‘*constitutional uptake*’ of DOP appears not to be widely used according to our literature search and conversations we have had with colleagues. However, the definition we found was used in the revised version of our manuscript (Dyhrman and Ruttenberg, 2006; Fitzsimons et al., 2020; Mahaffey et al., 2014; Sebastián et al., 2004).

According to this literature (see previous sentence), APA encompasses inducible APA that is sensitive to changes in environmental conditions (such as changing DIP concentrations) and constitutive APA that appears to represent the background level of APA (Sebastián et al., 2004) if we are not mistaken. If correct indeed, constitutive APA will not be sensitive to changing DIP concentrations, and therefore can be also detected under high DIP concentrations (Sebastián et al., 2004). However, APA under these conditions (e.g., eutrophic environments) usually is very low and therefore would not have a significant influence on our data (Extended Data Fig. 2) (Dyrhman and Ruttenberg, 2006; Mahaffey et al., 2014).

Nevertheless, to acknowledge this uncertainty, we have revised as follows:

“The activity of AP (APA) is mainly composed of inducible APA and constitutive APA. Although there is evidence of background APA at relatively low levels (constitutive APA) independent of DIP concentrations¹²⁻¹⁴, AP usually is induced by low concentrations of DIP. Thus, the APA is usually inversely correlated with DIP concentrations, and inducible AP therefore becomes activated for the utilization of DOP when DIP concentration falls below a certain DIP threshold¹⁵⁻¹⁷.”

Dyrhman, S. T., & Ruttenberg, K. C. (2006). Presence and regulation of alkaline phosphatase activity in eukaryotic phytoplankton from the coastal ocean: Implications for dissolved organic phosphorus remineralization. *Limnology and Oceanography*, 51(3), 1381-1390.

Fitzsimons, M. F., Probert, I., Gaillard, F., & Rees, A. P. (2020). Dissolved organic phosphorus uptake by marine phytoplankton is enhanced by the presence of dissolved organic nitrogen. *Journal of Experimental Marine Biology and Ecology*, 530, 151434.

Mahaffey, C., Reynolds, S., Davis, C. E., & Lohan, M. C. (2014). Alkaline phosphatase activity in the subtropical ocean: insights from nutrient, dust and trace metal addition experiments. *Frontiers in Marine Science*, 1, 73.

Sebastián, M., Arístegui, J., Montero, M. F., Escanez, J., & Niell, F. X. (2004). Alkaline phosphatase activity and its relationship to inorganic phosphorus in the transition zone of the North-western African upwelling system. *Progress in Oceanography*, 62(2-4), 131-150.

3. Line 69, Delete “furthermore”

Response:

Agreed. We have revised accordingly.

4. Lines 81–82, Delete sentence. Paragraph should start with “Recent studies in oligotrophic open oceans have revealed that....”

Response:

Agreed. We have revised accordingly.

5. Line 86, “Our previous work...”

Response:

Agreed. We have revised accordingly.

6. Line 88, Change “although” to “even though”

Response:

Agreed. We have revised accordingly.

7. Lines 96-97, “Consequently, P has now become the main limiting nutrient.

Response:

Agreed. We have revised accordingly.

8. Lines 109-110, “...in biogeochemical cycles of coastal seas.”

Response:

Agreed. We have revised accordingly.

9. Line 129, Rewrite to correct sentence structure.

Response:

Agreed. We have revised as follows:

“At Y3_{Spr} and B1_{Spr}, the Chl *a* concentrations were also high (3.07 – 4.47 $\mu\text{g}\cdot\text{L}^{-1}$), even though DIN ($< 1 \mu\text{M}$) and Si(OH)₄ (0.20 – 0.79 μM) concentrations were relatively low. This is likely because the seawater was collected during a spring bloom.”

10. Line 132, corresponded

Response:

Agreed. We have revised accordingly.

11. Line 133, Use the indefinite article.

Response:

Agreed. We have revised accordingly.

12. Line 152, “...irrespective of the limiting nutrient...”

Response:

Agreed. We have revised accordingly.

13. Lines 175-176, "...respectively, during the incubations."

Response:

Agreed. We have revised accordingly.

We have also found similar mistakes elsewhere and changed them as follows:

"where $Chl\ a_{c-avg}$ and $Chl\ a_{t-avg}$ stand for the average $Chl\ a$ concentration in the control and N, P, N+P, or aerosol treatment groups, respectively, over the time course during the microcosm experiments ($N = 3$)"

"where the ΔDIN and ΔDIP represent the change of DIN and DIP concentrations, respectively, during the incubations."

"DIP utilization (ΔDIP) and net DOP utilization (ΔDOP) are calculated during the incubations from the change of DIP and DOP concentrations, respectively."

14. Line 234, "...characterized as..."

Response:

Agreed. We have revised accordingly.

15. Line 250, This is interesting but what about constitutional uptake of DOP, which has been reported. This ties in with my first comment above.

Response:

Through the method of measuring APA, both the constitutive and inducible APA are captured. According to the general positive linear relationship between RC_{APA} and $\text{Log}_{10}(\text{Chl } a/\text{DIP})$ (Fig. 6a), as well as the increasing extent of APA (2% - 228%) induced by aerosol additions (Fig. 3), we conclude that the inducible uptake of DOP likely plays a dominant role in our study and not the uptake of DOP driven by constitutive APA.

Sebastián, M., Arístegui, J., Montero, M. F., Escanez, J., & Niell, F. X. (2004). Alkaline phosphatase activity and its relationship to inorganic phosphorus in the transition zone of the North-western African upwelling system. *Progress in Oceanography*, 62(2-4), 131-150.

16. Lines 263-267, I'm not clear about the relevance of novelty of this text; phytoplankton bloom in temperate waters and this study implies induced uptake of DOP.

Response:

What we have tried here to explain is that temperature can affect the overall rate of DOP utilization, but not so much its induction. Hence, temperature has an effect on the overall activity of the enzyme (e.g., Arrhenius equation) but DIP is responsible for its induction (e.g., initiation of overall activity

above background level), which is the main subject of our work. To make this clearer, we have amended our text as follows:

“The seawater temperature, although an important factor determining enzyme activity²⁰, is likely not the main determining factor for DOP utilization although it has an influence due to the temperature-dependence of enzyme activity. For example, in the subtropical Atlantic at relatively high seawater temperatures (21 – 29 °C) and off the coast of northern Oregon at relatively low seawater temperatures (11 – 12 °C), AP was activated regardless of the water temperature when phytoplankton experienced P stress^{13,17}.”

17. Line 317, Report the purity of the water; Milli-Q is a brand name.

Response:

Agreed. It was deionized water (18.2 MΩ cm).

18. Line 322, metal

Response:

Agreed. We have amended is accordingly.

19. Line 323 and throughout, Check that tenses are correct and consistent in the manuscript.

Response:

The tenses have been rectified throughout the manuscript.

20. Rees at al (2009) doi:10.1016/j.ecss.2008.12.005 observed an increase in APA in the Western English Channel and this corresponded with a salinity decrease related to increased riverine inputs, increasing the N:P ratio in the sampling area. If these inputs are riverine or atmospheric it stands to reason that DOP utilisation could be stimulated.

Response:

We have referred to this paper in our revised version and amended our text as follow:

“These results corroborate a study in the Western English Channel where the increase in APA was associated with increased river runoff that resulted in elevated N:P ratios¹⁸. Thus, apart from the atmospheric deposition, river runoff likely will also enhance DOP utilization in coastal seas through a similar mechanism. However, air pollutants relative to river runoff will have an impact over larger geographical areas because of atmospheric long-range transport³⁸.”

21. The last sentence on curbing nitrogen inputs is vague and could be constrained using estimates of anthropogenic inputs, such as Jickells et al (2017) doi:10.1002/2016GB005586. This could

strengthen Figure 5, where N deposition is key.

Response:

We have referred to this paper in our revised version and amended our text as follow:

“Atmospheric deposition is considered an important source of nutrients²²⁻²⁴ for the surface ocean, as it usually contains significant amounts of N and trace metals (e.g., Fe) but relatively low concentrations of P^{23,25,26}.”

“Our results suggest that DOP utilization by microorganisms such as phytoplankton will become more important in the future. Atmospheric N input to the ocean has significantly increased from 1850 (10 Tg N yr⁻¹) to 2005 (39 Tg N yr⁻¹) and is expected to remain elevated until at least 2100 (-17% - +8%)²⁴ at a business-as-usual scenario. Thus, the Atmospheric Nitrogen Pump is at least expected to have a continued impact on the biogeochemistry of coastal seas globally, and its role might even become more prominent under increasing N deposition²² unless we reduce the input of terrigenous N-rich material such as aerosol and riverine water.”

Reviewer #3:

General comments:

1. In my opinion, the authors fail to convincingly demonstrate 1) that DOP utilization is stimulated by aerosol addition, 2) that APA is stimulated by aerosol addition which would be required for DOP utilization, and 3) that the driving mechanism leading to enhanced DOP utilization by phytoplankton is the alteration of the ambient DIN:DIP by aerosol addition.

Response:

With all due respect, our data show that APA and DOP uptake are enhanced after the addition of aerosols (Fig.3, Fig. 5a, S8). As APA is responsible for DOP uptake, we have provided a causative relationship with our work driven by the addition of N-rich aerosols depleting the pool of DIP before APA is induced to utilize DOP. The latter is possible under these circumstances because there is sufficient DIN in the system (>> DIN:DIP) due to “The Atmospheric Nitrogen Pump”. To make this clearer, we have added the following text:

“These results corroborate a study in the Western English Channel where the increase in APA was associated with increased river runoff that resulted in elevated N:P ratio¹⁸. Thus, apart from the atmospheric deposition, river runoff likely will also enhance DOP utilization in the coastal seas through a similar mechanism. However, air pollutants relative to river runoff will have an impact over larger geographical areas because of atmospheric long-range transport³⁸.”

2. River water amendment microcosms were conducted at B1 with the magnitude of the DOP utilization effect comparable to that from aerosol addition, weakening the authors conclusion that aerosols alone are important for stimulating DOP utilization in coastal seas.

Response:

Both atmospheric deposition and riverine water have elevated N:P nutrient ratios and are therefore regarded as important terrestrial nutrient sources for coastal seas (Jickells et al., 2017; Kim et al., 2011; Peñuelas and Sardans, 2022). The impact of atmospheric deposition covers a broader range through long-range transport, while that of rivers is more concentrated in nearshore areas (Galloway et al., 2008; Li et al., 2022). In this study, however, we tested the role of riverine water in DOP because it is a known source of DIN. Hence, it served as a control for which we already knew the outcome, but it was not the focus of our work because there is much more literature on the role of riverine N-input in coastal seas compared to the role of atmospheric deposition. Furthermore, we did not say anywhere in our manuscript that aerosols ‘alone’ are important for stimulating DOP utilization in coastal seas. In fact, our work provides first evidence that they are an additional source of N with significant consequences for our understanding of P-cycling in coastal seas around the globe.

3. As acknowledged by the authors, due to the weak support from interpreting enhanced DOP utilization in the microcosms by differences in measured DOP concentrations, the authors present a derived metric, DOP*, to support the notion that DOP utilization is enhanced by aerosol addition. DOP* represents the theoretical potential maximum in DOP utilization that might have occurred in the microcosm after computing the residual between the expected phytoplankton P demand (computed from the observed DIN utilization and Redfield N:P stoichiometry) and the observed DIP utilization. Using DOP* as the metric to gauge stimulation of DOP utilization by aerosol addition, raises the number of stations exhibiting a positive effect from 2 out of 8 to 4 out of 8, still not a very conclusive result in my opinion. The APA data presented in Figure 3 generally support the interpretations from using delta DOP or delta DOP*; i.e. in microcosms where a stimulation effect was observed for aerosol addition on DOP utilization, APA is higher and vice versa.

Response:

Our microcosm experiments can be divided up into three types according to the main source of phosphorous:

I. DIP was the main source of phosphorous. At E1 and Y1-DCM, DIP was the main P nutrient in control and treatment groups.

II. Equal contribution of DIP and DOP (i.e., the transition from P repletion to deficiency). At Y3_{Spr} and Y2, DIP was the main P nutrient in the control group, and DOP utilization was induced and

even became the main P nutrient in our aerosol treatments.

III. DOP was the main source of phosphorous. At $B1_{Spr}$, $B1_{Sum}$, Y1, and $Y3_{Sum}$, DOP was the main P nutrient in our control and treatment groups.

In our study, ΔDOP^* can be activated by aerosol additions corresponding to II and III (see above) (i.e., $Y3_{Spr}$, $B1_{Spr}$, Y2, $Y3_{Sum}$, and $B1_{Sum}$), where P deficiency occurred (Figs. 4, S8).

To make this clearer, we amended the manuscript as follows:

“I. DIP was the main source of phosphorous. In experiments using eutrophic seawater with relatively high DIP concentrations (0.07 – 0.11 μM) including E1 and Y1-DCM, DIP was the main bioavailable P nutrient, and 0.04 - 0.05 μM DIP was consumed by phytoplankton in the control groups (Figs. 4a and S8a, f).”

“II. DIP and DOP were equally used as a source of phosphorous. DIP was also the main P nutrient at $Y3_{Spr}$ and Y2 (Fig. S8b, g).”

“III. DOP was the main source of phosphorous. At $B1_{Spr}$, $B1_{Sum}$, Y1, and $Y3_{Sum}$, where DIP was highly deficient (0.01 – 0.03 μM), DOP contributed 82.4% - 92.5% of P and became the primary P nutrient for phytoplankton growth in the control groups (Fig. 4c). However, although DOP was still the main P nutrient in the treatment groups, no significant enhancement of ΔDOP was found after aerosol additions (Fig. S8c, d, e, h).”

“At E1 and Y1-DCM, where DIP was the main P nutrient, ΔDOP^ was low (-0.06 – 0.02 μM) (Fig. 4a).”*

“In addition, ΔDOP^ significantly responded to aerosol additions at $Y3_{Spr}$, $B1_{Spr}$, Y2, $Y3_{Sum}$, and $B1_{Sum}$ where DOP was the main P nutrient or might have become the main P nutrient after aerosol additions, of which the response was positively correlated with the amount added (Figs. 4b-c and S8).”*

4. As for the conclusion that elevated DIN:DIP is responsible for elevated DOP utilization, the data in Figure 4b largely supports this by identifying a significant positive correlation between the relative APA stimulation (expressed as a percentage relative to the controls) versus DIN:DIP as the predictor variable. However, dissolved iron has also been identified as an important co-control on DOP utilization in marine surface waters (see Browning et al. 2017 Nat. Comm., Liang et al., 2022 Nat. Geo.). The authors need to demonstrate data and/or other arguments to include or rule out the potential control of enhanced DOP utilization by Fe addition from aerosols and possibly river water.

Response:

Dissolved iron controlling DOP utilization is primarily important in oligotrophic open oceans

(Browning et al. 2017; Liang et al. 2022), and we also referred to these two papers in our manuscript.

To provide further evidence, we have conducted **an additional microcosm experiment while we revised our manuscript** using surface seawater collected in the Yellow Sea (37.00 °N, 123.83 °E).

These additional results corroborate our finding that iron additions will not enhance APA in these coastal waters but nitrogen (Fig. S4a – see below). These new results have been added to the supplementary information (Fig. S4). Additional text has been added to the main manuscript as follow:

“Although the role of essential trace metals such as Zn and Co for the APA was not tested in our study, the APA did not significantly respond to Fe or N+Fe additions (Extend Data Fig. 1 and Fig. S4), and the APA in the aerosol treatment was comparable to those of the riverine water treatment (Figs. 3b-c and S5a-b), suggesting that other trace metals usually enriched in aerosols and riverine water were likely not having a substantial impact on our results.”

Fig. S4 | Response of activity of alkaline phosphatase (APA) to iron and nutrient additions. *a*, APA response in the microcosm experiment conducted in the laboratory using surface seawater collect in the Yellow Sea during summer 2023. *b*, *c*, APA response in the microcosm experiment conducted in the East China Sea (E2 Autumn) and the Yellow Sea (Y5 Autumn) during autumn 2020. Significant differences are indicated by different letters and NS = not significant (one-way ANOVA and Turkey HSD post hoc test). Error bars indicate standard deviation; N=3.

Harmesa, Wahyudi, A., Lestari, & Taufiqurrahman, E. . (2021). Variability of trace metals in coastal and estuary: distribution, profile, and drivers. Marine pollution bulletin, 174, 113173.

Karavoltos, S., Sakellari, A., Dassenakis, M., Bakeas, E., & Scoullou, M. (2021). Trace metals in the marine surface microlayer of coastal areas in the Aegean sea, Eastern Mediterranean. *Estuarine, Coastal and Shelf Science*, 259, 107462.

5. The authors should report the limit of detection and analytical uncertainty for DOP determinations after considering the additive uncertainties from the separate determinations of TDP and DIP. The precision on TDP analyses is unreported so it is impossible as a reviewer to determine if the delta DOP values reported in Figure 3 on the order of -30 to +90 nM are even larger than the analytical uncertainty on DOP determinations.

Response:

We have added the error margins of DIP (1.39% of the average value of 0.06 μM) and TDP (0.77% of the average value of 0.20 μM) analyses which were calculated by the relative standard deviation of five replicates. The detection limit for DOP analysis (0.025 μM) was calculated by the sum of those of DIP and TDP ($DL_{\text{DIP}} + DL_{\text{TDP}}$). The error margins for DOP analysis (1.27% of the average value of 0.14 μM) were calculated using the standard deviations (SD) and average values (avg) of five replicates for TDP and DIP analysis ($\frac{\sqrt{SD_{\text{TDP}}^2 + SD_{\text{DIP}}^2}}{TDP_{\text{avg}} - DIP_{\text{avg}}} \times 100\%$). Detailed modifies are showed as follows:

“The error margin for the DIP analysis was 1.39% of the average value of 0.06 μM , which represents the relative standard deviation of five replicates.”

“The detection limit of TDP was 0.02 μM , and the error margins for TDP analysis was 0.77% of the average value of 0.20 μM through the same calculating method as that of DIP”

“DOP was estimated by subtracting DIP from TDP. The detection limit (0.025 μM) for DOP analysis was calculated by the sum of those of DIP and TDP ($DL_{\text{DIP}} + DL_{\text{TDP}}$). The error margin for DOP analysis (1.27% of the average value of 0.14 μM) was calculated using the standard deviations (SD) and average values (avg) of five replicates for TDP and DIP analysis

($\frac{\sqrt{SD_{\text{TDP}}^2 + SD_{\text{DIP}}^2}}{TDP_{\text{avg}} - DIP_{\text{avg}}} \times 100\%$).”

Lastly, I am excited by the identified $\log_{10}(\text{chl}/\text{DIP})$ metric for predicting APA and potentially DOP utilization (Fig. 5). I believe many marine biogeochemists will find its predictions at the global scale to be useful in helping predict the role of DOP in the local to regional biogeochemistry across the global ocean.

Response:

Thank you very much. We are also excited about such relationship, because data about DOP utilization in the global coastal ocean is relatively scarce (Lin et al., 2016; Burt et al., 2018; Shao et al., 2023). Thus, we hope our study is a valuable contribution shedding new light on the role of atmospheric N deposition in biogeochemical cycles of coastal seas.

Burt, W. J., Westberry, T. K., Behrenfeld, M. J., Zeng, C., Izett, R. W., & Tortell, P. D. (2018). Carbon: Chlorophyll ratios and net primary productivity of Subarctic Pacific surface waters derived from autonomous shipboard sensors. *Global Biogeochemical Cycles*, 32(2), 267-288.

Lin, S., Litaker, R. W., & Sunda, W. G. (2016). Phosphorus physiological ecology and molecular mechanisms in marine phytoplankton. *Journal of Phycology*, 52(1), 10-36.

Shao, Z., Xu, Y., Wang, H., Luo, W., Wang, L., Huang, Y., ... & Luo, Y. W. (2023). Global oceanic diazotroph database version 2 and elevated estimate of global oceanic N₂ fixation. *Earth System Science Data*, 15(8), 3673-3709.

REVIEWER COMMENTS

Reviewer #1 (Remarks to the Author):

RE-REVIEW of a manuscript entitled "Atmospheric aerosols stimulate the utilization of dissolved organic phosphorus in coastal seas" by Haoyu Jin et al.

In the revised manuscript by Jin and his colleagues demonstrates comprehensive response to the concerns raised by both myself and another referee. I am satisfied with the revisions they have implemented., and I no longer have any reservations that would impede the publication of this paper. This manuscript offers valuable insights into the P cycle, with far-reaching implications for the carbon cycle in the northwestern Pacific.

Reviewer #3 (Remarks to the Author):

I appreciate the effort the authors have taken to consider my earlier concerns about the study. I am especially appreciative of the considerable extra effort involved to rapidly conduct another field study to specifically address the role of Fe nutritional status on APA and DOP utilization in their study area. I am now generally supportive of publication but not addition issues that need to be addressed by the authors.

1) Why do the authors compare the APA data to N^* in Fig. S10 and elsewhere and not P^* ? P^* as defined in Deutsch et al., 2007 Nature represents 'phosphorus excess' relative to nitrogen and Redfield phytoplankton requirements. Marine algae should increasingly turn to DOP to satisfy P demand as P^* decreases, which should drive up APA. Framing in terms of N^* or 'nitrogen excess' seems unintuitive to me. P^* has been used to infer DOP accumulation versus consumption across the open ocean before in this manner (Liang et al., 2022 Nature Geoscience). Also I don't believe that the authors have computed N^* correctly in the revised manuscript. N^* is defined as $N^* = [NO_3] - 16*[PO_4]$ when ignoring the scaling constants as introduced in Gruber & Sarmiento, 1997 GBC. The authors compute N^* as $([NO_3]/16) - [PO_4]$. I encourage revising of the use of N^* as a predictor variable in diagnosing APA or delta DOP patterns.

2) The new Figure 2 caption is lacking a description of subpanel b. It is also unclear to me from reading the caption and looking at the figure what the colored boxes represent. Are you presenting a box and whiskers plot? Why are the data points and the colored bars different colors within a station? Please consider revising the figure or its caption for clarity.

3) I am supportive of the reframing of the results into 3 groups of stations based on contributions of DIP and DOP to overall P demand and the associated Fig. 4. In fact it is this reframing that has helped be better interpret the study results and better convince me of the study's conclusions. I think reworking Fig. 3 to also group by P demand would also aid in interpretation.

4) Thank you for providing the key information regarding the measurement accuracy and precision for TDP, SRP, and DOP determinations.

5) My earlier comment concerning river vs aerosols additions as controls on stimulating DOP utilization and APA was motivated by the study's title. My interpretation of the title is that aerosol additions alone are the driving mechanism that alter the DIN:DIP ratio of coastal waters driving the system towards P limitation and thus DOP utilization. But your river amendment results and associated discussion in the text suggest that river inputs act in the same manner to alter the DIN:DIP. Thus I am supportive of a more inclusion title that highlights the role of aerosols and river inputs working in tandem in the coastal ocean to alter the DIN:DIP towards P limitation and DOP utilization.

Response to reviewers' comments. Text in *italics* (copied from revised manuscript) represents how we have addressed the reviewers' suggestions in our revised manuscript. Our response in blue.

Reviewer #1 (Remarks to the Author):

RE-REVIEW of a manuscript entitled "Atmospheric aerosols stimulate the utilization of dissolved organic phosphorus in coastal seas" by Haoyu Jin et al.

In the revised manuscript by Jin and his colleagues demonstrates comprehensive response to the concerns raised by both myself and another referee. I am satisfied with the revisions they have implemented., and I no longer have any reservations that would impede the publication of this paper. This manuscript offers valuable insights into the P cycle, with far-reaching implications for the carbon cycle in the northwestern Pacific.

Response:

Thank you very much for your positive response.

Reviewer #3 (Remarks to the Author):

I appreciate the effort the authors have taken to consider my earlier concerns about the study. I am especially appreciative of the considerable extra effort involved to rapidly conduct another field study to specifically address the role of Fe nutritional status on APA and DOP utilization in their study area. I am now generally supportive of publication but not addition issues that need to be addressed by the authors.

1) Why do the authors compare the APA data to N^* in Fig. S10 and elsewhere and not P^* ? P^* as defined in Deutsch et al., 2007 Nature represents 'phosphorus excess' relative to nitrogen and Redfield phytoplankton requirements. Marine algae should increasingly turn to DOP to satisfy P demand as P^* decreases, which should drive up APA. Framing in terms of N^* or 'nitrogen excess' seems unintuitive to me. P^* has been used to infer DOP accumulation versus consumption across the open ocean before in this manner (Liang et al., 2022 Nature Geoscience). Also I don't believe that the authors have computed N^* correctly in the revised manuscript. N^* is defined as $N^* = [NO_3] - 16*[PO_4]$ when ignoring the scaling constants as introduced in Gruber & Sarmiento, 1997 GBC. The authors compute N^* as $([NO_3]/16) - [PO_4]$. I encourage revising of the use of N^* as a predictor variable in diagnosing APA or delta DOP patterns.

Response:

Thank you for your additional comments to improve our work further. We calculated N^* because reviewer #1 suggested that in the first-round of revision. It tries to shed light on the impact of P deficiency on APA, based on the inverse relationship between APA and DIP concentrations. Upon recalculating N^* following your advice, we found that it still exhibits a weak correlation with the relative change of APA after aerosol addition (Fig. S10a). Similarly, we implemented the reviewer's advice to calculate P^* in terms of DOP accumulation or consumption, but found no significant correlation between the relative change of APA and P^* . Considering the well-established relationship between relative change of APA and $\text{Log}_{10}(\text{Chl } a/\text{DIP})$ in this study, the above results also support the important role of biomass (Chl *a*) in affecting DOP utilization. Therefore, we combined N^* and P^* in Fig. S10 and added the explanation as follows:

“In addition, the relative change of APA was not significantly correlated with N^ ($\text{DIN}-16 \times \text{DIP}$) in terms of P deficiency or P^* ($\text{DIP}-\text{DIN}/16$) in terms of DOP consumption or accumulation^{42,65} (Fig. S10), suggesting the important role of phytoplankton biomass (Chl *a*) in affecting DOP utilization in coastal seas”*

Fig. S10. Dissolved organic phosphorus utilization indicated by the activity of alkaline phosphatase (APA) and its correlation with N^* and P^* . Correlation between the relative change of APA after treatment and the parameters N^* ($\text{DIN}-16 \times \text{DIP}$) (a) and P^* ($\text{DIP}-\text{DIN}/16$) (b), of which APA_C and APA_T represent APA on the day they began to enhance (In the middle of incubations) in the control and aerosol/riverine water treatments, respectively. DIN and DIP represent the concentration of dissolved inorganic nitrogen and dissolved inorganic phosphorus, respectively. 16 represents the Redfield ratio of N:P. Positive N^* indicates DIP deficiency, and negative P^* indicates DOP consumption. The blue and red lines (best-fit line) with shades (95% confidence level) represent the results of the linear regression with and without data of E1 (covered by grey shades), respectively.

2) The new Figure 2 caption is lacking a description of subpanel b. It is also unclear to me from reading the caption and looking at the figure what the colored boxes represent. Are you presenting a box and whiskers plot? Why are the data points and the colored bars different colors within a

station? Please consider revising the figure or its caption for clarity.

Response:

We are presenting a box plot to show the limiting nutrient and the distribution of ΔR across different nutrient additions. The coloured dots indicate ΔR in different nutrient treatment groups, and the coloured boxes indicate the limiting nutrients (blue = N limitation, yellow = P limitation, red = N+P colimitation, and white = not significant). To make it clearer, we have modified Fig. 2 and its caption as follows:

Fig. 2 | Response of Chl a concentration to nutrient, aerosol and riverine water additions expressed by ΔR . ΔR ($\log_{10}(\text{Chl } a_{t\text{-avg}}/\text{Chl } a_{c\text{-avg}})/t$) is the Log_{10} ratio of the average Chl a concentration during the incubations in treatment relative to control groups. A) ΔR in nutrient addition experiments. The coloured dots indicate ΔR in different treatment groups. The colour in boxes indicates the limiting nutrients (blue = N limitation, yellow = P limitation, red = N+P colimitation, and white = not significant) ($p < 0.05$). The lines across the boxes represent 25th (bottom) and 75th (top) percentiles. The whisker caps represent minimum (bottom) and maximum (top) values. Significant level was calculated using one-way ANOVA and Turkey HSD post hoc test. B) ΔR in aerosol and riverine water addition experiments. The lines across the boxes represent

25th (bottom), 50th (middle), and 75th (top) percentiles. The whisker caps represent minimum (bottom) and maximum (top) values. Significant differences ($p < 0.05$) between ΔR in spring and summer are indicated with a connecting line and p value (unpaired t -test).

3) I am supportive of the reframing of the results into 3 groups of stations based on contributions of DIP and DOP to overall P demand and the associated Fig. 4. In fact it is this reframing that has helped be better interpret the study results and better convince me of the study's conclusions. I think reworking Fig. 3 to also group by P demand would also aid in interpretation.

Response:

We have revised Fig. 3 as follows:

Fig. 3 | The activity of alkaline phosphatase (APA) in response to aerosol addition. Significant differences between different groups at the same day of the incubations are indicated by different letters and NS = not significant (one-way ANOVA and Turkey HSD post

hoc test). Error bars indicate standard deviations; N=3.

4) My earlier comment concerning river vs aerosols additions as controls on stimulating DOP utilization and APA was motivated by the study's title. My interpretation of the title is that aerosol additions alone are the driving mechanism that alter the DIN:DIP ratio of coastal waters driving the system towards P limitation and thus DOP utilization. But your river amendment results and associated discussion in the text suggest that river inputs act in the same manner to alter the DIN:DIP. Thus I am supportive of a more inclusion title that highlights the role of aerosols and river inputs working in tandem in the coastal ocean to alter the DIN:DIP towards P limitation and DOP utilization.

Response:

We have changed “atmospheric nitrogen pump” to “anthropogenic nitrogen pump” and revised the manuscript wherever we did not mention the role of rivers. This includes the title, abstract, introduction, results, discussion, and references. We have also amended figure 7 to include the importance of rivers. Please see our version with tracked changes for all details and amendments. Many thanks for this suggestion. It makes sense and it broadens the reach of our paper.